# Mapping the landscape of histomorphological cancer phenotypes using self-supervised learning on unannotated pathology slides

Adalberto Claudio Quiros[1,2,13], Nicolas Coudray[3,4,5,13], Anna Yeaton[6], Xinyu Yang[1], Bojing Liu[6,7], Hortense Le[5,6], Luis Chiriboga[6], Afreen Karimkhan[6], Navneet Narula[6], David A. Moore[8,9], Christopher Y. Park[5], Harvey Pass[10], Andre L. Moreira[6], John Le Quesne[2,11,12,14] ✉, Aristotelis Tsirigos[3,5,6,14] ✉ & Ke Yuan[1,2,11,14] ✉

Cancer diagnosis and management depend upon the extraction of complex information from microscopy images by pathologists, which requires time-consuming expert interpretation prone to human bias. Supervised deep learning approaches have proven powerful, but are inherently limited by the cost and quality of annotations used for training. Therefore, we present Histomorphological Phenotype Learning, a self-supervised methodology requiring no labels and operating via the automatic discovery of discriminatory features in image tiles. Tiles are grouped into morphologically similar clusters which constitute an atlas of histomorphological phenotypes (HP-Atlas), revealing trajectories from benign to malignant tissue via inflammatory and reactive phenotypes. These clusters have distinct features which can be identified using orthogonal methods, linking histologic, molecular and clinical phenotypes. Applied to lung cancer, we show that they align closely with patient survival, with histopathologically recognised tumor types and growth patterns, and with transcriptomic measures of immunophenotype. These properties are maintained in a multi-cancer study.

Hematoxylin and eosin (H&E) stained tissue sections are the mainstay of cancer diagnosis and many treatment decisions. Their ubiquitousness makes them the single largest source of data for studying the highly heterogeneous phenotype of tumors, yielding information from subcellular resolution to tissue architecture and complex interactions within the tumor microenvironment[1-3]. However, pathologist interpretation is time-consuming and prone to inter-observer variation, depending on expertise, knowledge, and on the inherent difficulty in characterizing certain tumors or patterns[4-7]. Supervised deep learning methods have shown to be on par with specialists on tumor

classification tasks[8-10]. In addition, these approaches have also been employed to tackle more challenging questions, such as predicting genetic alterations[8,11,12], survival[13-15] and immunotherapy response[16].

While such approaches can lead to accurate models, obtaining rigorous clinical annotations is difficult. Annotations are, however, crucial to properly train supervised models or to further study the significance of certain histomorphologies, e.g. in Johannet et al.[16], immunotherapy response was predicted by selecting tumor regions over lymphocyte-rich or connective tissue regions. Furthermore, by limiting the study to annotated features, such approaches also limit

the potential discovery of new bio-markers. Finally, such approaches are often described as black-boxes, where interpretation and understanding of how decisions are taken by the network are often difficult, and may affect trust, limiting the ability to take well-informed treatment decisions[17].

Semi-supervised and weakly-supervised approaches have emerged to alleviate this bottleneck. They can learn from a small subset of labeled data, and have proven to be beneficial in various applications[18–20], including histopathology[21–25]. These methods range from a cluster-based approach on support vector machine for breast cancer tissue classification[26], to teacher-student architectures on large colorectal cancer datasets[27]. Multiple instance learning (MIL) naturally fits tasks for WSI label prediction[28]. Of particular relevance is the attention-deep MIL model[29], which introduces interpretability into a MIL deep learning model by providing attention scores to the WSI tiles and has been extensively used and adapted to histopathology[30–32]. However, MIL only informs about which individual tile(s) are important for a given task without giving any further information about the broader clinical and biological significance of the tiles.

Parallel to these methodology advancements, interest in unsupervised and self-supervised methods is growing in the field of histopathology[33–39]; unlike supervised approaches, these models create representations of tissue images without the need of labels and solely from information encapsulated in the image. This line of work has recently been applied to a range of different tasks, demonstrating for example, that tumor regions are not necessarily the best predictor for tumor mutations[35], that variational auto-encoders (VAEs) can disentangle morphological components of single cells from H&E stained images[40], or that self-supervised model can be successfully used for segmentation of cell nuclei[41].

Here, we propose an unbiased method to extract histomorphological phenotype representations through self-supervised learning and community detection. Besides the self-discovery of histomorphological phenotypes, our approach provides a mechanism to link Histomorphological Phenotype Clusters (HPCs) to clinical and molecular annotations, without the need to retrain the model as would be required by supervised and weakly-supervised end-to-end solutions. In addition, our methodology is interpretable, allowing pathologists to scrutinize tissue patterns and their relations to annotations such as cancer type, overall survival, recurrence-free survival, or molecular phenotypes, providing therefore phenotype-to-molecular-to-clinical associations. To illustrate our framework, we first applied it to the analysis of whole-slide images of lung adenocarcinoma (LUAD), a cancer with many sub-types and heterogeneous features, where tumor morphology is highly predictive of patient outcome. We first show how the clusters obtained with our self-supervised pipeline effectively catalogue the highly diverse morphologies which comprise this tumor type, thus generating an atlas of histomorphologic phenotypes (HP-Atlas). We then demonstrate their clinical relevance, showing how they can be used to predict overall and recurrence-free survival, and our method's ability to identify, ab initio, tumor regions enriched for recognised cell types, growth patterns, and omic-based immune signatures. We then expanded our study to multicancer analysis, showing how histomorphologic patterns enriched in specific molecular features can be used to either distinguish cancer subtypes, such as lung adenocarcinoma from squamous cell carcinoma or be used to identify universal cancer phenotypes that predict overall survival in the context of a multi-cancer analysis.

## Results

### HPL: histomorphological Phenotype Learning through self-supervised learning and community detection

HPL discovers histomorphological phenotypes (HPs), i.e. distinct morphological tissue patterns, in large collections of whole-slide images (WSI) by employing an unbiased, self-supervised deep learning approach in which training does not require any label or annotations to be provided by expert pathologists. Once these distinct patterns are identified, any new WSI obtained from a patient sample can be characterized by the patterns it contains. This allows expert pathologists to assess quantitatively the composition of new samples in terms of histomorphological patterns.

Figure 1 outlines the entire HPL methodology. HPL uses a dataset of WSIs as input, without any annotations by experts. First, WSIs are segmented into 224 × 224 tiles without overlap. Tiles are filtered out if the tissue in the image does not cover at least 60% of the area. Then, stain normalization is applied[42] (Fig. 1A).

After initial pre-processing of WSIs into tiles, HPL uses self-supervised learning on a training set of tiles (see Methods for details) to capture distinct morphological patterns found in tissue and to represent them by vector representations, which can be thought of as feature vectors that describe the visually distinct patterns (e.g. texture). Through this process, each 224 × 224 tile image is transformed into a tile vector representation $\{z \in R^D; D = 128\}$. Importantly, HPL ensures that representations are invariant to color and slight zoom distortions, in order to mitigate the impact of variations in the imaging processing across institutions[43] (Fig. 1B). HPL uses Barlow Twins[44] for self-supervised learning, as this method has proven to be on par or to outperform other self-supervised methods, while eliminating the need for higher computational requirements such as large batches[45] or architecture asymmetries[46,47], thus providing competitive state-of-the-art results with significantly fewer resources.

Next, HPL defines a nearest neighbor graph between tiles, using the tile vector representations from the previous step, motivated by the idea that neighboring tile vector representations hold similar morphological features. Then, HPL uses Leiden community detection[48] on the nearest neighbor graph in order to find Histomorphological Phenotype Clusters (HPCs). In order to select the number of HPCs, we developed a self-supervised method that trades off HPC compactness and their ability to generalize across cohorts from different institutions (see Methods for details). The resulting HPCs are clusters of 224 × 224 tissue tiles that contain common morphological patterns (Fig. 1C).

Defining HPCs allows us to easily describe a single WSI or a patient (composed of one or more WSIs) by the frequency of these HPCs (Fig. 1D). More specifically, each WSI or patient is converted into a compositional vector ($w$) which has a dimensionality equal to the total number of HPCs ($C$) and each dimension ($w_i$) accounts for the percentage of area covered by each HPC with respect to the total tissue area (Equation (1)):

$$w = \{w_0, w_1, ..., w_{C-1}\} \text{ where } \{w_i \in [0, 1] / \sum_{i=0}^{C-1} w_i = 1\} \qquad (1)$$

While clustering as been used in previous deep-learning pipelines for tasks like image retrieval[49], tissue classification or tumor stage classification[50], no strategy for in-depth histopathological, molecular and clinical characterization was provided. As we will show in the subsequent sections, HPL's approach to quantifying WSIs and patients provides (1) interpretability through tissue patterns that can be easily visualized and analyzed in the context of WSI and by individual tiles, (2) the application of interpretable models such as logistic regression or Cox regression, verifying the statistical significance of phenotypes in diagnosis (e.g. cancer type classification) and clinical outcomes (e.g. survival), (3) broader association studies with genomic, transcriptomic and other multi-omic profiles, and (4) a measure of spatial heterogeneity as well as potential relationships between patterns.

In the following sections, we demonstrate the various applications of the proposed method. First, we study HPL's ability to identify meaningful histomorphological phenotypes in lung adenocarcinoma patient slides. Second, we characterize the identified HPCs using

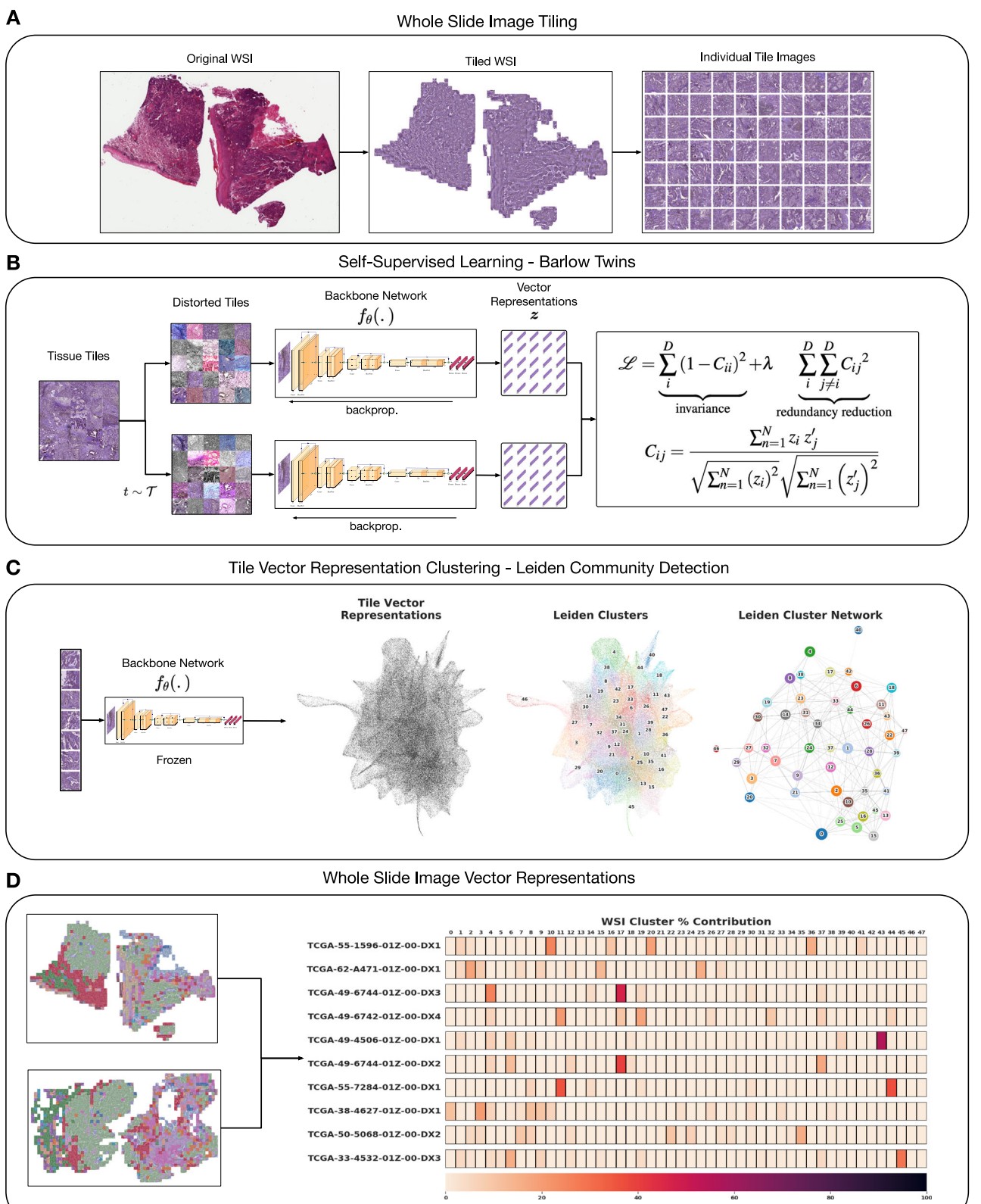

**Fig. 1 | Overview of Histomorphological Phenotype Learning (HPL) framework architecture. A** Whole slide images (WSIs) are processed for tile extraction and stain normalization. **B** The self-supervised training of backbone network $f_\theta$ creates tile vector representations. **C** Tiles are projected into $z$ vector representations using the frozen backbone network $f_\theta$. Continuously, Histomorphological Phenotype Clusters (HPCs) are defined using Leiden community detection over a nearest neighbor graph of $z$ tile vector representations. **D** WSIs or patients (one or more WSIs per patient) are defined by a compositional vector with dimensionality equal to the number of HPCs and accounts for the percentage of a HPC with respect to the total tissue area. **HPL** creates WSI and patient compositional vector representations that can be easily used in interpretable models such as logistic regression or cox regression, relating tissue phenotypes with clinical annotations. Source data are provided as a Source Data file.

clinical variables, cell type densities, and genomic and transcriptomic features. Third, we show how this pipeline can be used to predict in an interpretable way the overall and recurrence-free survival. Fourth, we identify trans-cancer features linked to molecular phenotypes and clinical outcomes in a multi-cancer setting.

## De novo mapping of the landscape of histomorphological phenotypes in lung adenocarcinoma

By design, HPL identifies visually similar patterns in order to organize input images (tiles) into distinct clusters (HPCs). First, we show that these distinct clusters of visually similar patterns correspond to meaningful histomorphological phenotypes. To this end, we focused on lung adenocarcinoma, a cancer type that has been shown to consist of an oftentimes complex mix of diverse histologic patterns (notably lepidic/in situ, acinar, papillary, cribriform, solid, and micropapillary), which are strongly linked to clinical outcomes[1].

We applied HPL to whole-slide images of lung adenocarcinomas obtained from TCGA. The slides were first split into tiles at 5x magnification, 224 × 224 pixels ($2\mu m/px$) in size. Tiles with more than 40% of background were filtered out (considering background pixels those which average grey-level is above 230 - Note that this will result in removal of some tiles at the edge of the tissue as well as some loose tissue regions). First, we sub-sampled 250,000K tiles for the self-supervised task. Second, we projected on the trained network all of the 432,231 tiles from the 541 slides corresponding to 452 patients affected by lung adenocarcinoma only. Then, 22,658 of those tiles were removed by a first clustering steps as they were visually identified as being artefacts (see Methods for details). Next, we performed dimensionality reduction on the remaining ~411,000 tile representation vectors using the Uniform Manifold Approximation and Projection (UMAP) method[51] (Fig. 2A). We then used Leiden community detection with an unsupervised resolution selection strategy (see Methods for details and Supplementary Fig. 1A), which resulted into identifying 46 HPCs (Fig. 2A). The strategy suggested here aims to find the minimal number of clusters that provides the most reproducible of features present across institutions. Other tasks or datasets may require using different resolution selection strategies.

To confirm that our data augmentation strategy (Fig. 1B) mitigated the technical biases from slide to slide or institution to institution due to staining, scanning and other factors, we tested whether the identified clusters are patient-specific or institution-specific. The number of patients who contribute tiles in each cluster is shown in Fig. 2B, demonstrating that none of the identified clusters is patient-specific, and all clusters but the last one (HPC 45) receive contributing tiles from at least 17%(78/452) patients. Overall, the HPCs are highly recurrent, appearing in a median of 32%(133/452) of all patients. Similarly, Fig. 2C shows the number of institutions that contribute to each cluster: no cluster was found to be institution specific, and all clusters but the last one contain tiles from at least 24%(8/33) institutions.

Then, to evaluate whether these 46 clusters correspond to meaningful histomorphological phenotypes, we randomly selected 100 tiles from each cluster and asked three expert pathologists to evaluate and annotate the tiles in terms of a number of characteristics: (1) detailed tissue morphology (first and second most predominant growth pattern/nontumor tissue component, depending on epithelial content); this was simplified into a consensus tissue category, (2) area ratio of epithelium versus stroma, and (3) the degree of lymphocytic infiltration (Fig. 2D). Using these annotations of representative tiles, we first colored the clusters in the UMAP by consensus histological appearances (Fig. 2E). We observed a clear separation by tissue constituents, suggesting that HPL is successful at capturing visual characteristics in the different tissue types. For example, tiles with malignant cells are concentrated on the left half of the UMAP, clearly separated from normal lung (bottom right) and specialised tissues

(upper right). Assessments of malignant epithelium:stroma ratio (Fig. 2F) and lymphocytic infiltration (Fig. 2G) also revealed striking zonation. In particular, within the UMAP area dominated by malignancy, there is a gradient between the more peripheral clusters, which have a higher tumor:stroma ratio (Fig. 2F) and fewer tumor infiltrating lymphocytes (TILs) than clusters situated more centrally (Fig. 2G).

For each cluster, a consensus summary title was derived from pathologist descriptions, and representative tiles for each cluster are shown in Supplementary Fig. 2. The degree of histopathological heterogeneity within clusters is variable but dominant features were generally discernable with good agreement between pathologists (Fig. 2D). Most clusters clearly recapitulate nameable, recognisable histopathological phenomena.

Looking at how these HPCs are grouped on the UMAP, we observe that the lower right quadrant contains HPCs populated by tiles with preserved alveolar architecture (Fig. 3A). Interestingly, two morphological trajectories are apparent: moving upwards from normal parenchyma (HPC 4) there is a transition through mild interstitial fibrosis/expansion (HPCs 32/13) to dense interstitial chronic inflammation (HPC 36). Alternatively, moving to the right, there is a progressive loss of air spaces due to iatrogenic compression (HPC 2) and then haemorrhage (HPC 10).

Several clusters in the upper right area of the UMAP represent specialised non-organoid appearances (eg necrosis, confluent inflammatory cells) and non-malignant mesenchymal tissues seen in the background of tumor resection tissue blocks (Fig. 3B): cartilage, bronchi, vessels, confluent lymphocytic inflammation and necrosis all form well-defined clusters in this area. More centrally there are numerous clusters dominated by different types of stroma (Fig. 3B, left) and stroma-rich tumor, with greater or lesser degrees of inflammation, and different qualities of extracellular matrix. The remaining clusters mostly define various recognisable manifestations of adenocarcinoma (Fig. 4A). Clusters characterised by all 6 classical patterns are identified. Lepidic-enriched clusters also contained numerous tiles histopathologically judged to be acinar or papillary in nature, highlighting current difficulties in histopathological identification of low-grade invasive disease[52]. The 'lepidic' cluster with thinner septa is close to mucinous disease in the UMAP (HPC 28), while the other, with bulkier interstitium, is close to reactive and inflamed normal lung clusters (HPC 37). Remaining examples of clusters of disease with clear growth pattern associations are situated in distinct neighbourhoods of the UMAP. Interestingly, the most dedifferentiated HPCs (i.e. solid pattern) lie at the opposite pole of the UMAP to the most differentiated lepidic and mucinous clusters.

A further interesting group of clusters show variants of classical growth patterns, (Fig. 4B): HPCs 6, 11, 27 and 39 are characterised by different variants of solid growth with clefts, holes and fragmentation suggestive of discohesion, as has been described recently[53]. All are situated in a 'grey area' that lies between classical cribriform and solid patterns. HPC 12 is a rather diverse group of mixed metaplastic features and low-grade malignancy, while HPC 33 appears to be largely defined by retraction of solid nests from thin septa. Two further clusters are defined by artefacts: HPC 44 contains tissue folds, and HPC 17 is made up of tiles with large empty areas due to pleural surfaces or other tissue edges (Supplementary 5).

Taken together, these results show that, given a large collection of whole-slide images, and without any expert labels or annotations, the proposed self-supervised pipeline constructs an HP-Atlas of meaningful and distinct histomorphological phenotypes. HPL was able to identify all previously described histologic patterns in lung adenocarcinoma, and, in addition, generate distinct clusters for complex mixed patterns, while also capturing different inflammatory patterns, structural lung features and non-malignant lung tissues that have been affected by a diverse set of processes, from fibrosis to haemorrhage.

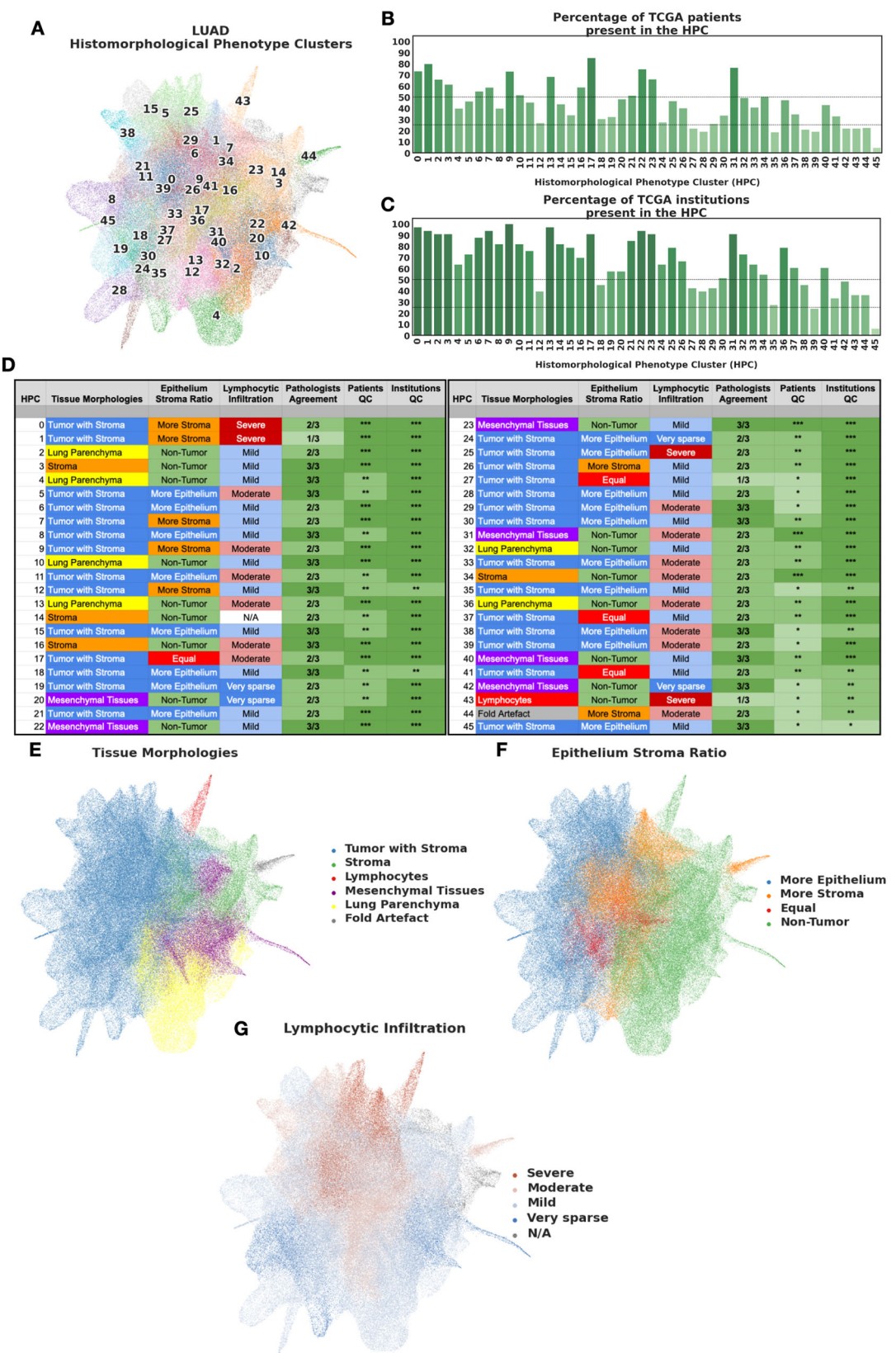

HPL-identified HPCs can then be used to quickly inspect whole-slide images and appreciate the spatial heterogeneity that may be present in patient samples. In Fig. 5 we show examples of whole-slide images obtained from three representative TCGA patients. Figure 5A corresponds to TCGA-80-5608 who was lost to follow-up (i.e. alive) seven years after surgery, Fig. 5B corresponds to TCGA-38-4625 who

was lost to follow-up (i.e. alive) eight years after surgery, and Fig. 5C that corresponds to patient TCGA-50-5931 who died 14 months after surgery. For each case, we display the original H&E image (top) and a modified version ("HPC-map") of the same image overlaid by a distinct color for each HPC (bottom). In this manner, it is easy to visually explore not only the heterogeneity of a whole slide image, but also the

**Fig. 2 | HPCs from Lung adenocarcinoma show consistent enrichment in histomorphological phenotypes. A** Uniform Manifold Approximation and Projection (UMAP) dimensionality reduction of lung adenocarcinoma tile vector representations labeled by HPC membership (each HPC was assigned a different color for easier visualization). **B** Percentage of patients from the TCGA cohorts associated with each HPC (100% corresponding to 452 patients). The shades of green are proportional to the percentages (y-axis). **C** Percentage of institutions associated with each HPC (100% corresponding to 33 institutions). The shades of green are proportional to the percentages (y-axis). **D** Consensus annotations of each HPC after visual inspection by a panel of 3 expert pathologist of 100 random tiles from each HPC. Stars for detailed consensus indicate the number of agreeing pathologists for the predominant tissue component (a given growth pattern/ non-tumor element, see details in Methods - Cluster Histological Assessment), while the number of stars for patients and institutions quality control (QC), are related to panels B and C with percentage above 50%, above or below 25% for 3, 2 and 1 star respectively). Labels were then projected back to the UMAP in panels E-G. Visual representations of **E** the distribution of the different tissue categories, **F** the epithelium:stroma ratio, and **G** the extent of lymphocytic infiltration are displayed on the UMAP. Source data are provided as a Source Data file.

spatial relationships of the different HPCs. Figure 5A shows a lepidic pattern mucinous adenocarcinoma with focal mucin pooling and peripheral reactive changes. Figure 5B is a predominantly solid pattern tumor with extensive lymphocytic infiltration, and multifocal cribriform appearances. Figure 5C is also predominantly solid pattern, but crucially shows little evidence of TIL infiltration. There are central zones of necrosis, and entrapped vessels and airways are also highlighted.

### HPL identifies prognostic histomorphological phenotypes in lung adenocarcinoma associated with clinical outcomes

Next, we asked whether those HPCs can be used to model clinical outcomes in LUAD patients. We defined patient vector representations of the contribution of each HPC as the ratio of the area covered by the HPC to the total tissue area of all slides from the same patient. We then tested the HPCs relevance in predicting overall and recurrence-free survival by using patient vector representations on Cox proportional hazards models.

For the LUAD overall survival analysis, we used a 5-fold cross-validation over the TCGA data with a cohort from NYU ($NYU_1$) as an additional independent set. The TCGA cohort is composed of 443 patients, while $NYU_1$ is composed of 276 patients. We used a Cox proportional hazards model over patient vector representations, describing each patient as a composition of the different HPCs. HPCs are able to achieve a mean concordance index (c-index) of 0.60 with 95% confidence intervals (CI) of 0.56–0.63 on the TCGA test set and a mean c-index of 0.65 with 95% CI of 0.63–0.67 on $NYU_1$ (Supplementary Fig. 6A,B). Training Barlow-Twins on a subset of only 250,000 tiles did not seem to alter results, since similar results were obtained when running on the full training set (0.60 with 95% confidence intervals (CI) of 0.56–0.65 on the TCGA test set, 0.67 with 95% confidence intervals (CI) of 0.66-0.68 on $NYU_1$ Supplementary Fig. 7A). Similarly, HPL performs equally well when run at a higher magnification of 20x (0.61 with 95% confidence intervals (CI) of 0.57–0.64 on the TCGA test set, and 0.64 (CI of 0.63–0.64 on $NYU_1$, Supplementary Fig. 7B). Results also show robustness relative to the choice of the Leiden resolution, showing that over-clustering is not affecting the performance (Supplementary Fig. 7A,B). These results are comparable with the state-of-the-art supervised approaches (see Discussion). Supplementary Fig. 6C shows a SHAP (SHapley Additive exPlanations) plot of the log hazard ratio of the Cox model. It provides insight into the relationship between cluster phenotypes and their relevance in predicting overall survival; HPCs with high contributions (in red) and a positive SHAP value (top of the graph) are those associated with poor survival, while those at the bottom of the graph are HPCs associated with a predicted better survival. The Forest plot in Supplementary Fig. 8 provides an alternate analysis which provides hazard ratios but lead to comparable conclusions. In Supplementary Fig. 6D we show samples of the top HPCs associated with higher risk, and Supplementary Fig. 6E samples of HPCs associated with lower risk. Additional examples of tiles associated with HPCs are shown in Supplementary Fig. 9, where we can also see the coherence between the TCGA dataset and the $NYU_1$ external cohort.

We then conducted a recurrence-free survival study on systemic and loco-regional recurrence in LUAD. In this case, we used $NYU_1$ as it provides detailed information of these types of recurrence and has considerably longer follow up times compared to the TCGA dataset, allowing a more refined study of recurrence. Once again we used a 5-fold cross-validation, achieving a mean c-index of 0.74 with 95% CI of 0.70–0.80 on $NYU_1$ (Fig. 6A,B). As before, the pipeline is robust to changes in resolutions (Supplementary Fig. 7C). The SHAP (SHapley Additive exPlanations) plot (Fig. 6C) and the Forest plot (Supplementary Fig. 10) of the log hazard ratios help interpreting the predictions made by the HPL, showing at the top the HPCs with high contributions to favoring recurrence, while the presence of tiles from HPCs located at the bottom weighs against recurrence in the prediction model. Examination of tiles from the pro-recurrence HPCs reveals solid pattern malignancy with few lymphocytes, and often a degree of discohesion (HPCs 11, 15, 6, Fig. 6D). Anti-recurrence HPCs are inflamed and reactive in appearance, with low-grade tumor growth patterns (HPCs 36, 32, 37, Fig. 6E; more examples of tiles associated with HPCs in Supplementary Fig. 11).

In Fig. 6F, we illustrate this SHAP interpretation with a patient who eventually recurred 37 months after surgery, and was properly identified as high risk for recurrence at the time of surgery by HPL. We show the proportion of tiles (right column) associated with each cluster (left column) and how the proportion or absence of certain HPC contributes to the final prediction. HPCs are ordered, such as those contributing the most to the SHAP value are at the top. In this example, we see that the most important phenotypes are the presence of immunologically mildly inflamed solid pattern growth (HPC 15) and solid pattern disease with stroma-confined TILs (HPC 5), and the absence of the fibrotic lung pattern with lymphocytic infiltration associated with HPC 32. Such an approach is used to quantify the contribution of each HPC in the Cox regression model, which can be used by pathologists to gain an insight into how the model assigns risk to each HPC per patient.

### Systematic association of HPL-discovered patterns with cell types, histological growth patterns, and molecular phenotypes

In addition to providing a direct visual interpretation of relevant clusters of tiles, HPL allows us to quantitatively characterize HPCs in three different ways: (1) by using Spearman's rank correlation between cluster contributions and transcriptomic-based immune signatures such as tumor infiltrating leukocytes (TIL), proliferation, or wound healing[54], (2) by annotating cell types in the tiles from the $NYU_1$ cohort with Hover-Net[55], and using the two-sample Kolmogorov-Smirnov test to measure over and under-representation of cell types in each HPC, and (3) using representative manual annotations of regions at a slide level by pathologists in $NYU_1$, measuring enrichment or depletion of LUAD histological subtypes such as solid, acinar, papillary, micro-papillary, and lepidic using the hypergeometric test.

Through these characterisations we are able to provide further interpretation of prognostic HPCs, linking them to transcriptomic signatures, image-derived cell type counts and independently obtained pathologist annotations of growth pattern (Fig. 7A–C). Column dendrograms correspond to the bi-hierarchical clustering of

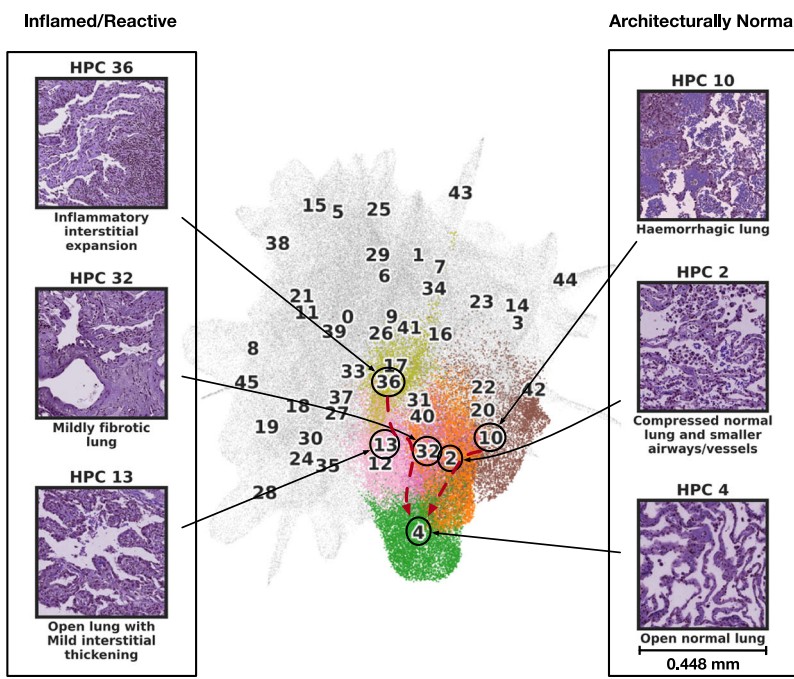

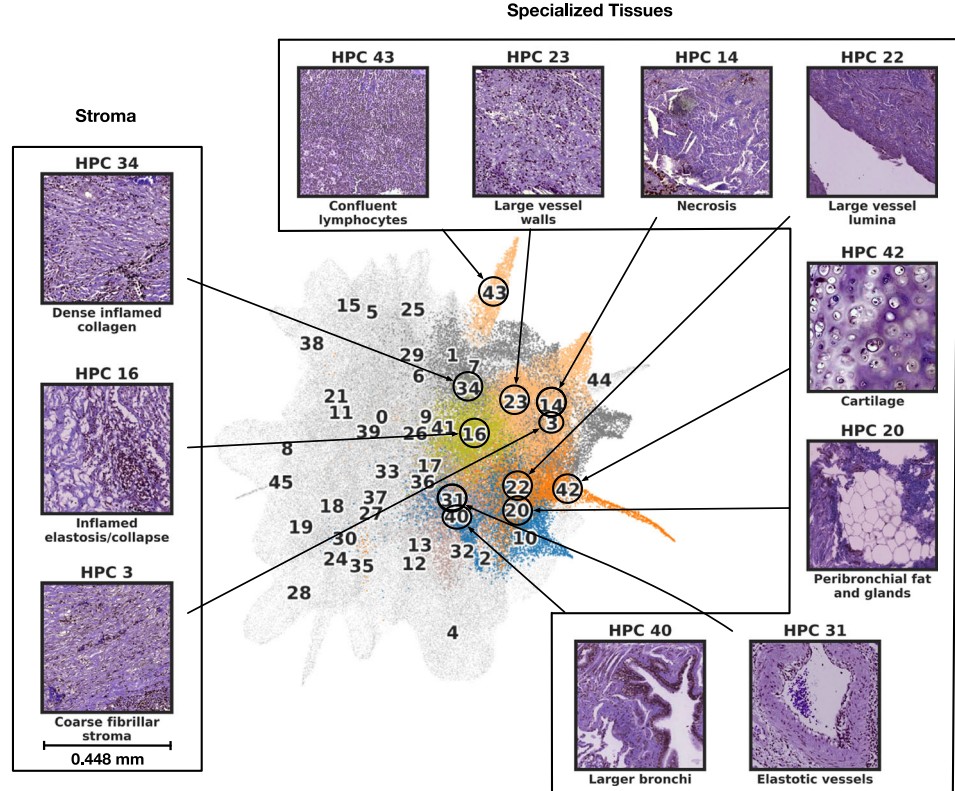

**Fig. 3 | Consensus description of HPCs enriched in nontumor phenotypes with their representative tiles. A** HPCs enriched with normal and reactive parenchyma. **B** HPCs enriched with stroma and other specialized tissues. We highlight tile vector representations of HPCs for each nontumor phenotypes **A** and **B**. HPCs of interest are colored as in Fig. 2A, while others HPCs remain grey. Consensus was obtained after independent annotations of HPCs by 3 pathologists as described in the Methods section - Cluster Histological Assessment. More examples of tiles for each HPC can be seen in Supplementary Figs. 3-4. Source data are provided as a Source Data file.

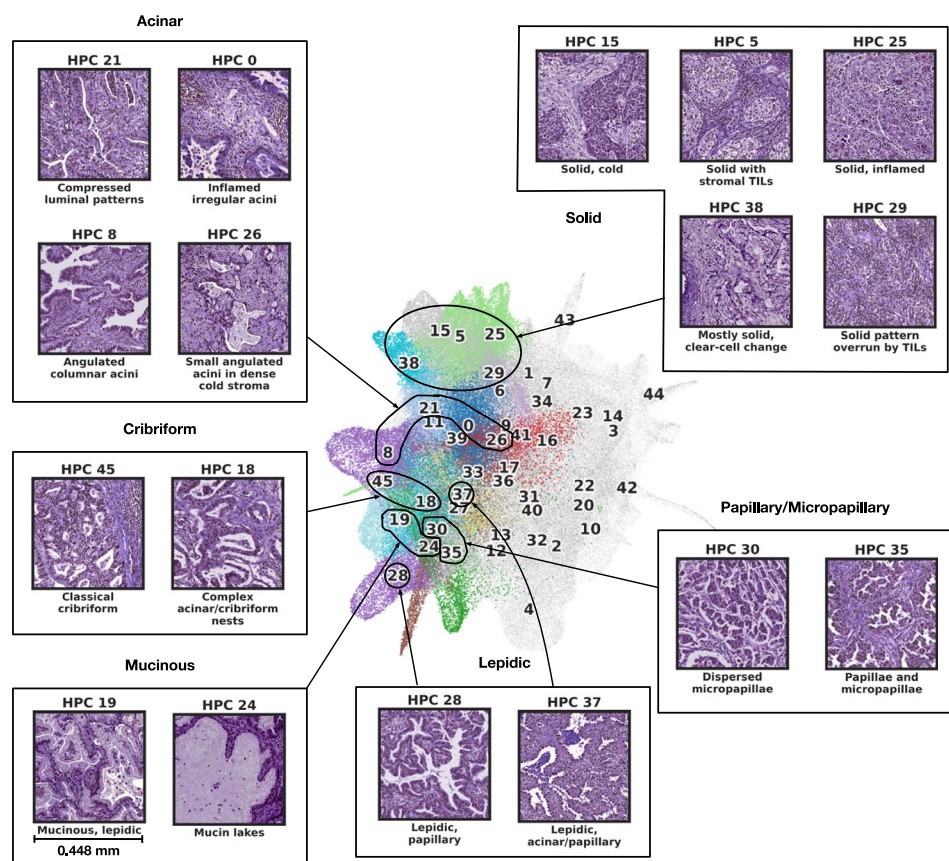

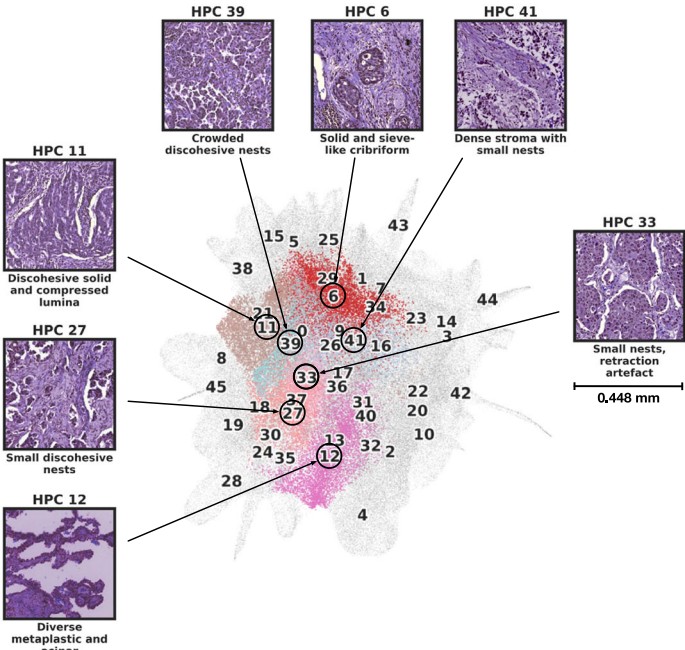

**Fig. 4 | Consensus description of HPCs enriched in tumor phenotypes with their representative tiles. A** HPC enriched with classical adenocarcinoma appearances. **B** HPCs enriched in variant adenocarcinoma appearances. HPCs of interest are colored as in Fig. 2A, while others HPCs remain grey. Consensus was obtained after independent annotations of HPCs by 3 pathologists as described in the Methods section - Cluster Histological Assessment. More examples of tiles for each HPC can be seen in Supplementary Figs. 3-4. Source data are provided as a Source Data file.

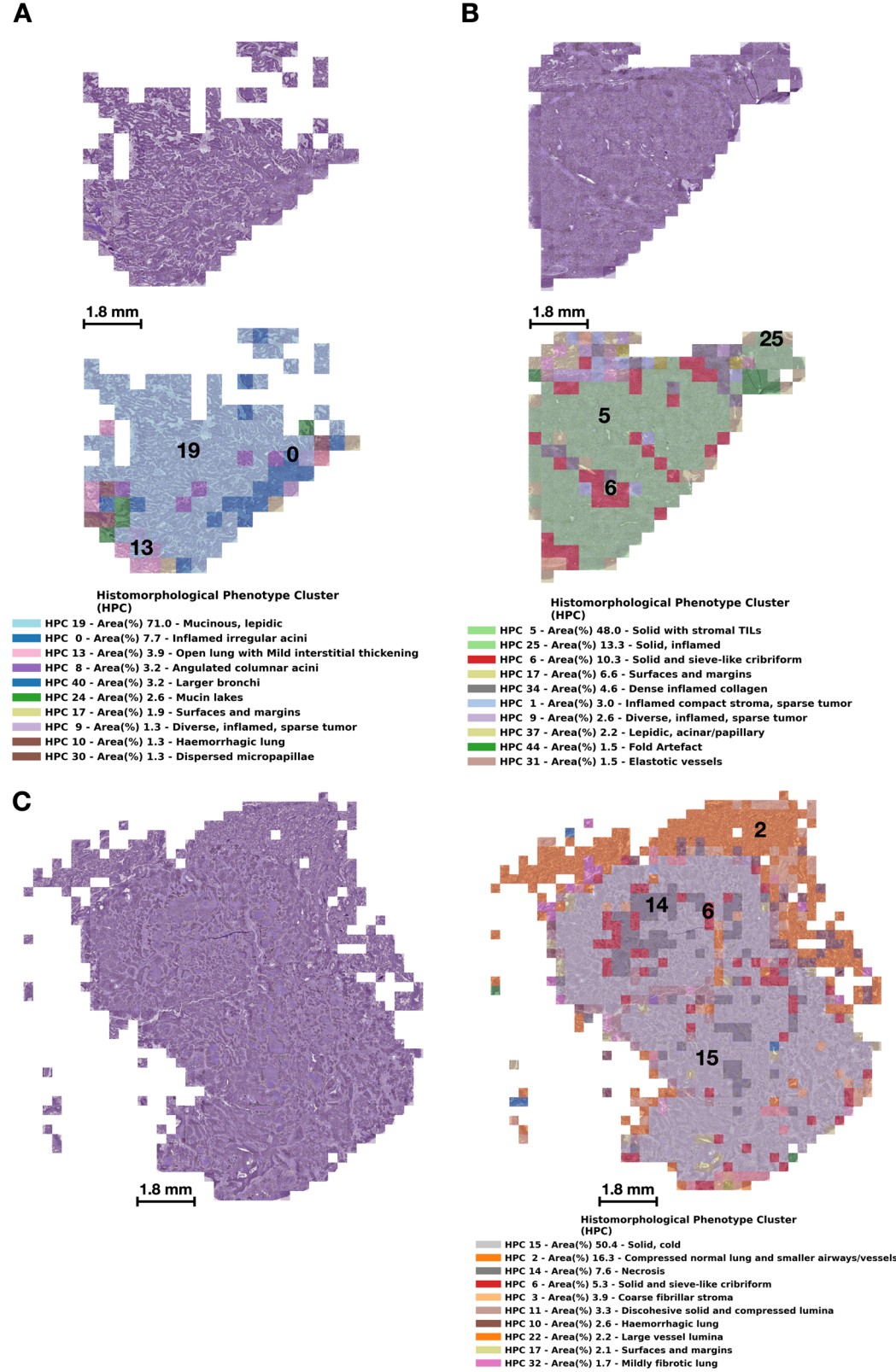

**Fig. 5 | Wholes slide images of lung adenocarcinoma with HPC overlays.** We display tumors from three representative TCGA patients. **A** corresponds to patient TCGA-80-5608 who was censored at a 7 year follow-up time, **B** corresponds to patient TCGA-38-4625 who was censored at a 8 year follow-up time, and **C** corresponds to patient TCGA-50-5931 who died 14 months after surgery. For each patient we show the original tile images (including tiles with at most 60% of background), and the same tiles but overlaid with a color code representing HPCs, and a legend with the percentage of tiles assigned to a given HPC; we display the most prevalent 10 HPCs per patient. Source data are provided as a Source Data file.

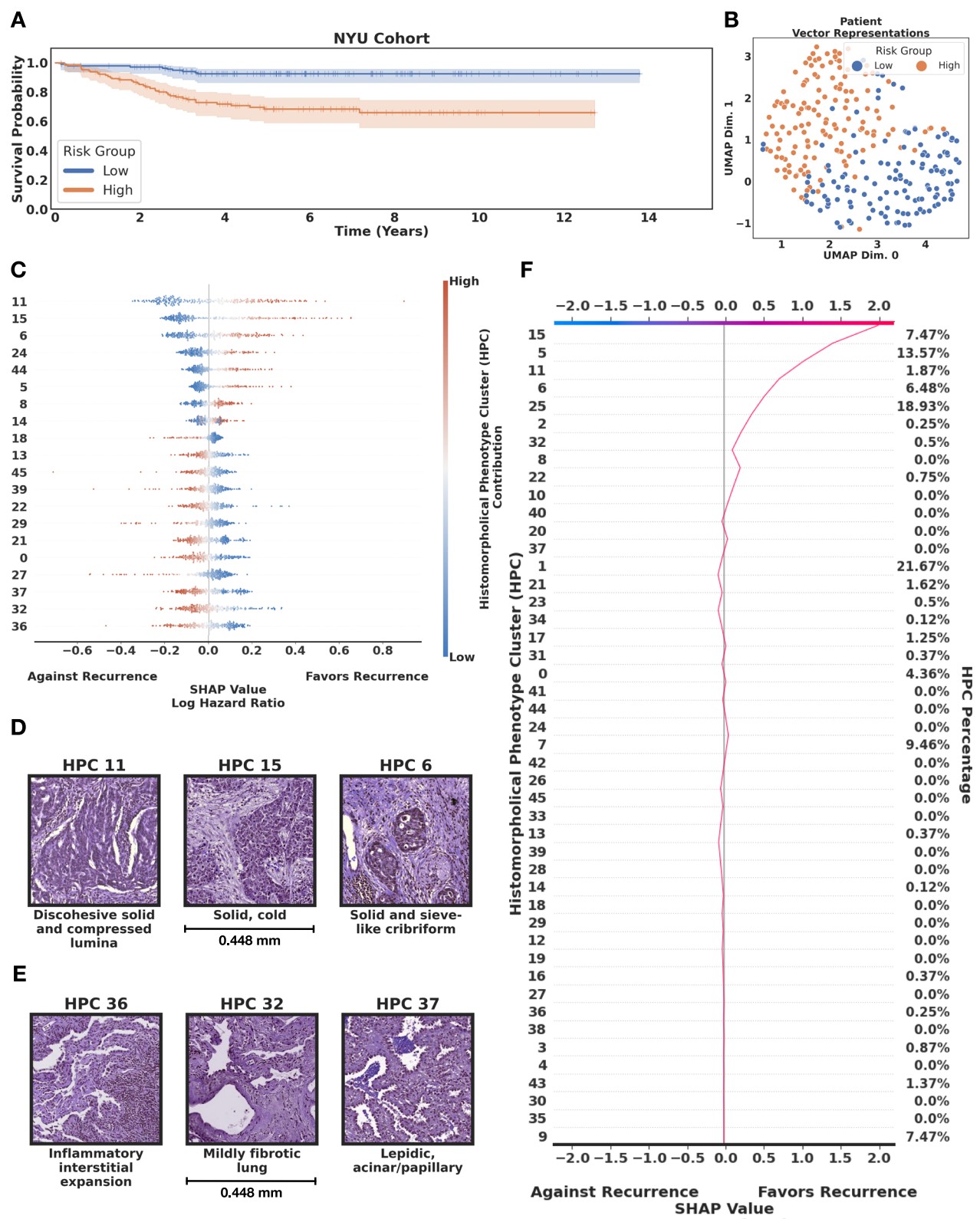

**Fig. 6 | Lung adenocarcinoma (LUAD) recurrence-free survival analysis by HPL.**
**A** High and low risk groups showing statistical significance ($p$ value $7.26 \times 10^{-6} < 0.05$ using the Logrank test). For each fold in the 5-fold cross validation we defined the high and low risk group threshold by taking the median risk value of the train set and we divided the test set into high and low risk based on this value. Since the test sets are non-overlapping across the 5-fold, at the end of the cross-validation all samples had been labeled as high or low risk based on the test sets of each fold. Error bars on the survival plots represent the 95% CI. **B** Uniform Manifold Approximation and

Projection (UMAP) dimensionality reduction of patient vector representations for the NYU cohort, each representation is labeled according to the risk group for recurrence, low-risk (blue) and high-risk (orange). **C** SHAP (SHapley Additive exPlanations) plot. **D** Top relevant HPCs associated with high risk of recurrence. **E** Top relevant HPCs associated with lower risk of recurrence. **F** Example of a decision plot for a patient slide classified as high risk of recurrence. We focus on HPCs that contain at least 10% of the total patients motivated by finding tissue patterns that can generalize across the cohort. Source data are provided as a Source Data file.

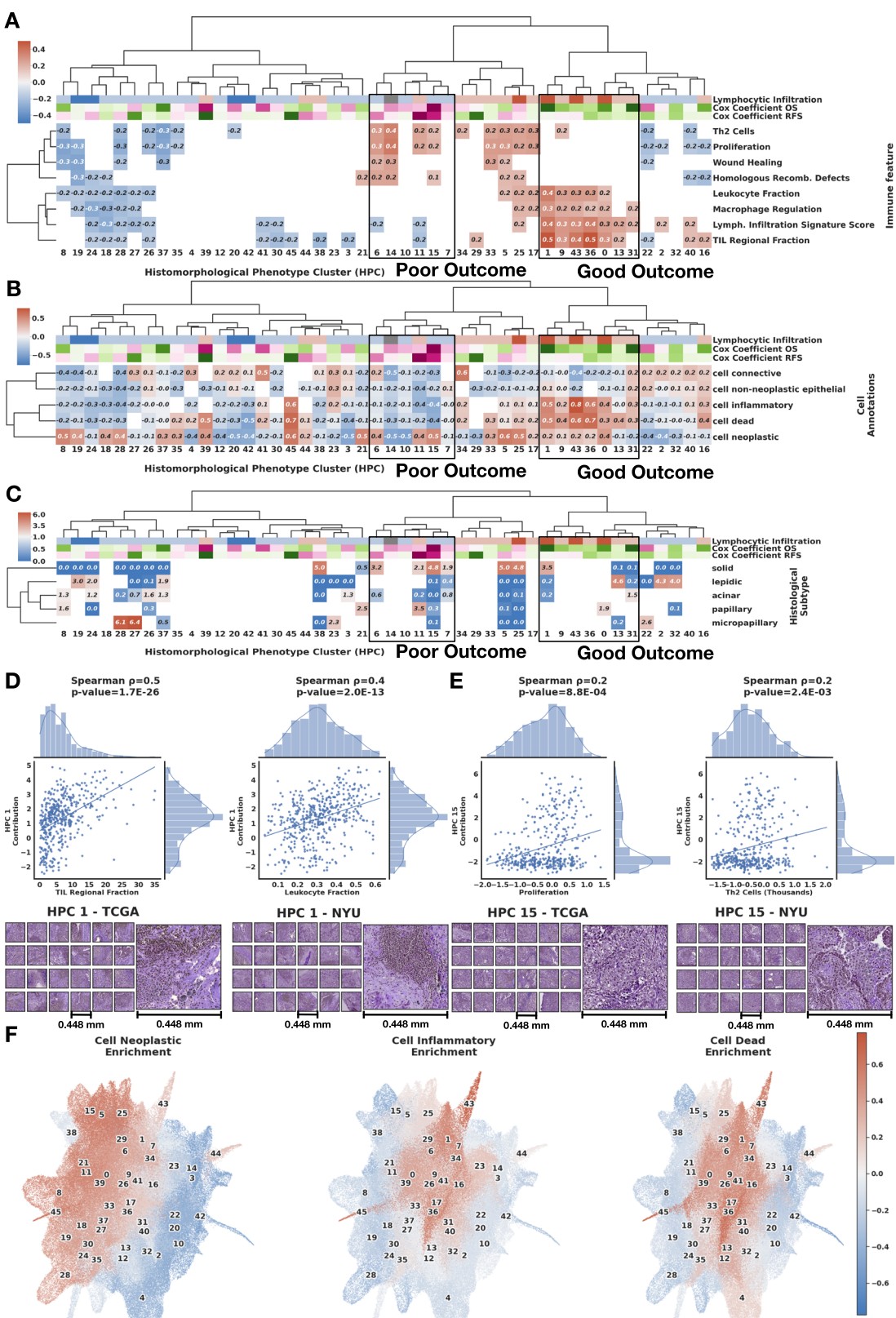

HPCs and immune signatures to relate these analyses in the same context. The bi-hierarchical clustering in Fig. 7A shows HPCs (tile contribution) and selected RNASeq-derived immune signature (obtained at a slide or patient level) correlations where positive correlations are shown in red and negative as blue. The trend shown at such resolution (5x magnification) are confirmed when HPL is run on higher resolution tiles (20x magnification, Supplementary Fig. 7D). We

expect correlations values could be even higher should ground truth labels be available at a tile level (from spatial transcriptomics, for instance). Again, analysis over a wide range of Leiden clusters show that over-clustering do induce over-fitting (although increasing the number of clusters increases the workload on the pathologist to annotate and review them), and unsurprisingly, it is only when the resolution is significantly lower (and features important for the

**Fig. 7 | Lung adenocarcinoma (LUAD) survival analysis and Histomorphological Phenotype Cluster (HPC) correlations. A** Bi-hierarchical clustering of HPCs and immune signature[54] with correlations from red (positive correlations) to blue (negative). Cox coefficients for overall and recurrence-free survival are colored from purple (favoring death or recurrence) to green (favoring survival or no recurrence). HPCs are colored based on histological assessment of lymphocytic infiltration: dark red: enrichment in severe infiltration; light red: moderate infiltration; light blue: mild infiltration; dark blue: very sparse infiltration; grey: other HPCs. **B** Bi-hierarchical clustering of HPCs and cell type over (red) and under-representations (blue). **C** Bi-hierarchical clustering of HPCs and LUAD histological subtype enrichment (red) or depletion (blue). For all panels, the column dendrogram in all subfigures corresponds to the bi-hierarchical clustering of HPCs and immune signatures to more easily relate these analyses in the same context. HPCs associated with poor and good outcome and associated hazard ratios (top rows) come from the Cox regression analysis shown in Supplementary Fig. 6 and Fig. 6 (see Methods - Cluster characterizations). HPCs associated with better survival outcomes show positive correlations with being severe to moderate lymphocytic infiltration and RNASeq signatures of tumor infiltrating leukocytes (TIL), lymphocyte infiltration signature score, T-cell receptors (TCR), and macrophage regulation; and show over-representations of inflammatory, dead, and neoplastic cells. HPCs associated with worse survival outcomes contain mostly mild lymphocytic infiltration content and show positive correlations with proliferation, mutation rate, homologous recombination defects, and wound healing signatures, under-representations for inflammatory and dead cells, and enrichment for solid histological patterns. **D** Scatter plot between HPC 1 contribution and omic-based immune signatures of each patient (tumor infiltration leukocytes (TIL) and leukocyte fraction), with representative HPC 1 tiles from TCGA and $NYU_1$ cohorts. **E** Scatter plot between HPC 15 contribution and omic-based immune signatures of each patient (proliferation and Th2 cells), with representative HPC 15 tiles from TCGA and $NYU_1$ cohorts. Two-sided Spearman correlation used for pannels D and E. **F** Uniform Manifold Approximation and Projection dimensionality reduction of the vector representations of the 224 × 224 tissue tiles, each tile label corresponds to the cluster cell type enrichment. Source data are provided as a Source Data file.

classification task likely grouped in a single HPC) that we start losing the ability to perform proper classification. The full set of signatures tested are made available in Supplementary Fig. 12. Figure 7B displays a bi-hierarchical clustering of over-representation (in red) or under-representation (in blue) of HPCs in certain cell types. Finally, Fig. 7C shows a bi-hierarchical clustering of HPCs and enrichment for an independent on-slide assessment of LUAD histological growth patterns; depletion is shown in blue and enrichment in red. Most HPCs are positively associated with a single annotation. Some show a mismatch between consensus title and annotation enrichment; in addition to the known poor inter-observer consensus in pattern assignation, this is usually explainable by the inclusion of stromal areas in areas of whole-slide annotation (eg HPC 3, 'fibrillar stroma', is enriched for acinar growth annotation), or by pattern adjacency/overlap blurring annotation accuracy (eg HPC 13, 'open lung with interstitial thickening' is enriched for lepidic annotation), or by pattern similarity (eg HPC 21 'compressed luminal patterns' is enriched for papillary annotation).

Throughout this analysis we found that HPCs that are associated with better outcomes are histologically enriched for lymphocytic infiltration, and have positive correlations with multiple molecular signatures reflecting inflammatory cells, and TIL and macrophage regulation along with neoplastic and dead cells, highlighting successful immune infiltration and elimination of tumor cells. On the other hand, HPCs that are associated with worse outcomes contain mild to very sparse lymphocytic infiltration and show positive molecular correlations with proliferation, mutation rate, homologous recombination defects and wound healing signatures, while showing under-representation of inflammatory and dead cells, and enrichment for solid histological patterns.

In addition, Fig. 7D, E provide further insight into representative HPCs linked to good outcome (HPC 1, "inflamed compact stroma, sparse tumor")) and poor outcome (HPC 15, "solid, cold"). Figure 7D displays scatter plots of each patient's contribution from HPC 1 and its relationship with transcriptomic assessments of TIL and leukocyte fractions, with examples of corresponding individual tiles from TCGA and $NYU_1$ cohorts. Similarly, Fig. 7E shows scatter plots of each patient's contribution from HPC 15 and its relationship with proliferation and Th2 cells, as well as examples of corresponding individual tiles from TCGA and $NYU_1$ cohorts. These results show the regional consistency of sets of HPCs associated with specific enrichment or depletion of certain cell types.

Finally, we further analyzed the relationship between HPCs with respect to over and under-represented cell types. Figure 7F shows a UMAP of the vector representations of the 224 × 224 tissue tiles, where each tile's label is defined by its enrichment in a certain cell type. These three figures highlight how similar HPCs show enrichment or depletion in the same cell types, also revealing how different parts of the space defined by the tile representations capture information about the density of neoplastic, inflammatory, and dead cells.

## Multi-cancer application of HPL unveils HPCs associated with cancer sub-types and clinical outcomes

Next, we explored how HPL can assist in identifying and interpreting HPCs across cancer types. First, we validated our approach assessing HPL's ability to distinguish lung adenocarcinoma (LUAD) from lung squamous cell (LUSC), a question well explored by supervised algorithms which can therefore be used as a positive control. We performed all the steps using TCGA as a training set, and used the New York University ($NYU_2$) cohort as an independent test set.

We used a logistic regression over WSI vector representations to perform LUAD vs LUSC classification, setting up a 5-fold cross-validation over TCGA's whole slides images (WSIs) and using $NYU_2$ as an independent test set. Each TCGA fold is built from a training, validation, and test set with WSIs from different institutions across sets. On average, the training set accounts for 60% of the TCGA slides, validation for 20%, and test set for 20%, while the NYU slides are held out as an external test cohort. The motivation behind setting a 5-fold cross-validation with nonoverlapping institutions is to test HPL reliability against the variability of slide preparation and imaging processing across institutions. HPL achieved an average TCGA test set AUC of 0.930 with 95% confidence intervals (CI) of 0.913–0.949, and an average of 0.990 with 95% CI of 0.988-0.992 on the independent test cohort from NYU. Since the tumor content on the NYU dataset is quite low (around 50% on average), it shows the algorithm is not affected by the high content of tissues not relevant for the LUAD/LUSC classification. Furthermore, whenever necessary, the pathologist' diagnosis was supplemented by immunohistochemical stains (TTF-1 and p40 for LUAD and LUSC, respectively), which may also result in a cleaner dataset with perfectly trustable labels and explain the high performance on that external dataset. These performances are also robust relative to the selection of the Leiden resolution, and over-clustering (higher Leiden resolution) would result in similarly good performances (Supplementary Fig. 17). Unsurprisingly, we observe a decrease of performances at lower Leiden resolutions (leading to fewer HPCs), when phenotypes crucial for differentiating LUAD from LUSC become grouped in more general HPCs. These results provide evidence of robustness in the prediction of LUAD and LUSC types across different institutions (Supplementary Fig. 20A-B). While the UMAP clearly shows regions enriched in tiles from LUAD or LUSC cases (Supplementary Fig. 20D), we also can visually confirm that the statistically significant HPCs contain features characteristic of either LUAD or LUSC (Supplementary Figs. 18, 19). On the TCGA dataset, we provide further results on AUC performance per institution and institution contribution per HPC on Supplementary Figs. 20, 22. One of the main features

of HPL's approach is the interpretability behind each prediction by describing which HPCs contributed the most to the classification: Supplementary Fig. 20 D–F provides insight into the relationship between histomorphological phenotypes and classification of WSIs into LUAD and LUSC. Finally, we performed uncertainty quantification through label-conditioned conformal prediction[56] (Supplementary Fig. 21, finding higher uncertainty in TCGA LUAD samples compared to TCGA LUSC ones. HPL displays similar confidence on LUAD and LUSC samples of the NYU cohort, with higher uncertainty in LUSC compared to TCGA. Nonetheless, HPL makes larger correct single predictions on both subtypes in the NYU cohort than in TCGA.

Next, we applied HPL to a multi-cancer dataset from TCGA (Fig. 8A), for which we selected cancer types with at least 150 samples and with immune signature information available from the GDC data portal. The motivation behind this study was the identification of common HPCs across multiple cancer types, and their association with immune features to understand if there are consistent links between phenotypes, molecular profiles and patient outcomes, therefore using criteria based on a curve (Supplementary Fig. 1C) which balances compactness (Equation 4) and representativity. Since we are interested in analyzing common features regardless of the cancer type and across institutions, it makes sense that the more diverse the dataset, the lower the number of common features (34 HPCs in this case). Figure 8B shows HPL identifies clusters enriched or depleted in tiles from samples associated with signatures of TIL density ("regional fraction"), macrophage regulation, proliferation, or TGF-beta response across cancer types. When displayed on the UMAP of the tile vector representation (Fig. 8C-H), we see how different types of enrichment and depletion are localised. While HPCs associated with TIL regional fraction occupy part of the left side of the UMAP (Fig. 8C) and partially overlap with those associated with proliferation and wound healing (Fig. 8D, G), we can see that HPCs associated with TGF-beta response mostly lies on the right-hand side of the UMAP, overlapping with stromal signature-enriched HPCs (Fig. 8E). This interesting observation concurs with known roles for TGF-beta signalling in stromal fibrosis and immune cell exclusion/suppression[57]. At the top of the UMAP graph, macrophage regulation and stromal fraction HPCs overlap in different ways with the TIL regional fraction and TGF-beta response HPCs. A comprehensive Spearman's rank correlation analysis of immune feature correlation with the HPCs is shown in Fig. 9A, where only statistically significant correlations are highlighted.

We then investigated how these multi-cancer phenotypes relate to overall survival. Initial review of Fig. 9A shows how clusters associated with good outcome are mostly enriched in immune transcriptional phenotypes such as pan-lymphocytes or CD8+ T-cells. Inspection of tile images shows dense lymphocytic infiltrates overrunning islands of malignant epithelium (HPCs 10, 14, 13, Supplementary Fig. 24) or inflamed stroma (HPC 8). The second group of HPCs associated with good outcome without molecular evidence of lymphocytic infiltration are more diverse, but in carcinomas are typically characterised by well-differentiated glandular/organoid appearances (HPCs 27, 28, 30, Supplementary Fig. 25). Poor outcome HPCs also fall into two groups: one is linked to markers of proliferation and genomic alteration, with histopathological appearances of poorly differentiated and infiltrative growth (HPCs 0, 20, Supplementary Fig. 27), while the other is linked to immune signatures in combination with responses to TGF-beta, and often show further solid and infiltrative appearances (HPCs 24, 15, 23, Supplementary Fig. 26), in keeping with established roles for TGF-beta signalling in epithelial-mesenchymal transition and invasiveness in later biological stages of malignancy[58]. In Fig. 9B, we show a 5-fold cross-validation per cancer type analysis, with values above 0.5 (in red) showing HPCs statistically significantly correlated with death (poor outcome), while values below 0.5 (in blue) show clusters associated to survival (good outcome). We can see, for example, that lung, breast,

skin cancers, and, to a lesser degree, bladder and colon share several HPCs which tile enrichment is related to the outcome. Taken together, these findings underscore the fact that even in disease as biologically diverse as melanoma and adenocarcinomas of diverse tissue origins, some morphological principles dominate cancer risk; brisk immune cell infiltration is a widely generalizable feature of relatively positive outcome, just as architectural dedifferentiation and infiltrative growth are ominous. In Table 1 (first column), we summarize for each cancer type the highest and lowest c-index associated with each cancer type. Further details of the mean and 95% confidence intervals of the C-index values over the 5-fold cross-validations are included in Supplementary Fig. 23.

## Discussion

We propose Histomorphological Phenotype Learning (HPL) as a self-supervised methodology for de novo identification of histomorphological phenotypes in whole slide images (WSI). HPL consists of four major steps, (1) WSI pre-processing, (2) self-supervised learning of tissue tiles, (3) tissue tile representation clustering into Histomorphological Phenotype Clusters (HPCs), and (4) HPC characterization. This approach therefore performs learning and clustering of tiles from histopathology WSIs in an unbiased manner, removing the need for expensive annotations, and providing clear insights in the decision process for a selected task.

Our HPL methodology was applied to identify and study histomorphological phenotypes specifically associated with lung adenocarcinoma. Crucially, HPL's HPCs recapitulate established LUAD growth patterns, with each World Health Organization (WHO) described growth pattern, as well as the mucinous subtype, predominating in at least one cluster, and usually more than one with different levels of lymphocyte infiltration (Fig. 2). No fewer than 7 clusters are predominantly solid growth patterns, being differentiated by the presence of stromal bands, lymphocyte infiltration, cytological appearances, cohesiveness, necrosis and folding artefacts. Other HPCs align closely with nontumor phenomena, such as bronchial cartilage, vascular smooth muscle, haemorrhage, and metaplastic changes in architecturally normal lung. Cluster contributions were examined for their ability to predict overall survival and recurrence-free survival. Our results show that lymphocyte density (measured both by histological appearances and by association with signatures from bulk RNASeq data) in tumor regions is highly relevant for overall survival prediction. TIL density is well established as being of prognostic and predictive value in other common malignancies (e.g. breast cancer[59]), colorectal cancer[60]. Our wholly self-supervised discovery of the positive outcome associations of TIL infiltration in lung adenocarcinoma lends strong support to previous histopathological studies linking TIL density to good outcome in NSCLC (review by Hendry et al.[61]), and more recent supervised deep learning studies of lung cancer WSIs and gene expression data[30,62]. For the overall survival prediction, our method achieved a mean concordance-index (c-index) of 0.65 over the external NYU cohort ($NYU_1$), and of 0.60 over TCGA only using WSIs, as compared to *Chen et al.* where the authors used histological and gene expression data achieving a mean c-index of 0.59, or 0.548 only using histological data. There have also been numerous studies using methods to link survival to single features[63,64]. In Lu et al.[65] for example, a neural network is used to detect the epithelial region and then segment the nuclei within it. In the subsequent analysis of nuclear image features, the authors identified 23 features, such as eccentricity and solidity or intensity measurements with prognostic value. In Wang et al.[66], the authors show the overall shape of the tumor also has some predictive value (hazard ratio of 2.25). HPL also highlighted that clusters associated with worse outcomes contain more classically high-risk growth patterns, while the presence of more predominant low-grade lepidic growth favors low risk. This mirrors the current practice in which histological grade is estimated by the pathologist from growth

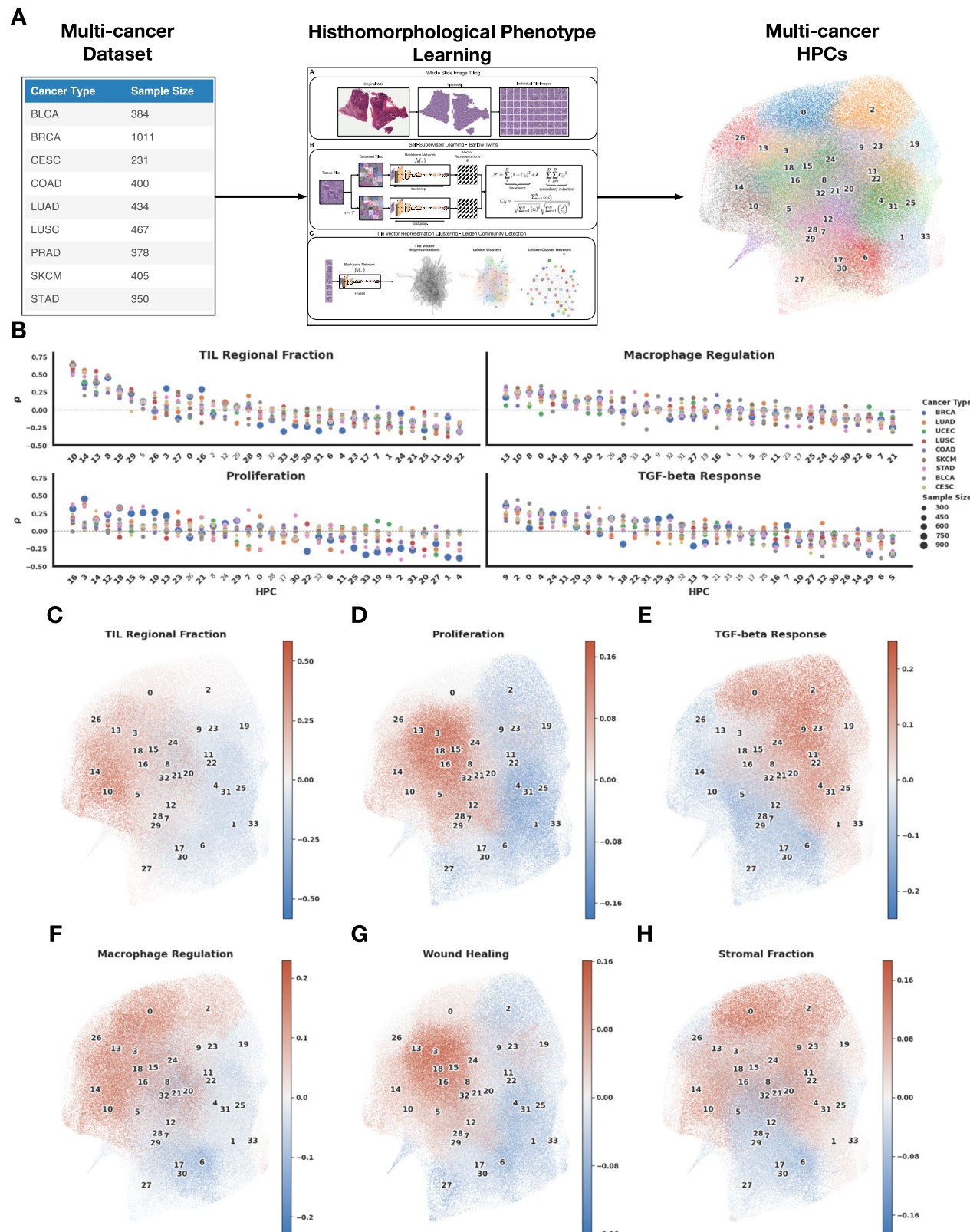

**Fig. 8 | Multi-cancer HPL pipeline and main enrichments of the resulting HPCs.**
**A** Multicancer pipeline: the 10 selected cancer types with sample sizes from 232 to 1011 patients (left) were fed to the HPL pipeline (middle), leading to 34 HPCs (right). **B** Example of 4 transcriptomic immune features which were highly correlated with specific HPCs as identified through Spearman correlation, and visualization on the UMAPs of tiles from HPCs highly enriched (in red) and depleted (in blue) in **C** TIL regional Fraction, **D** proliferation, **E** TGF-beta response, **F** macrophage regulation, **G** wound healing and **H** stromal fraction. Source data are provided as a Source Data file.

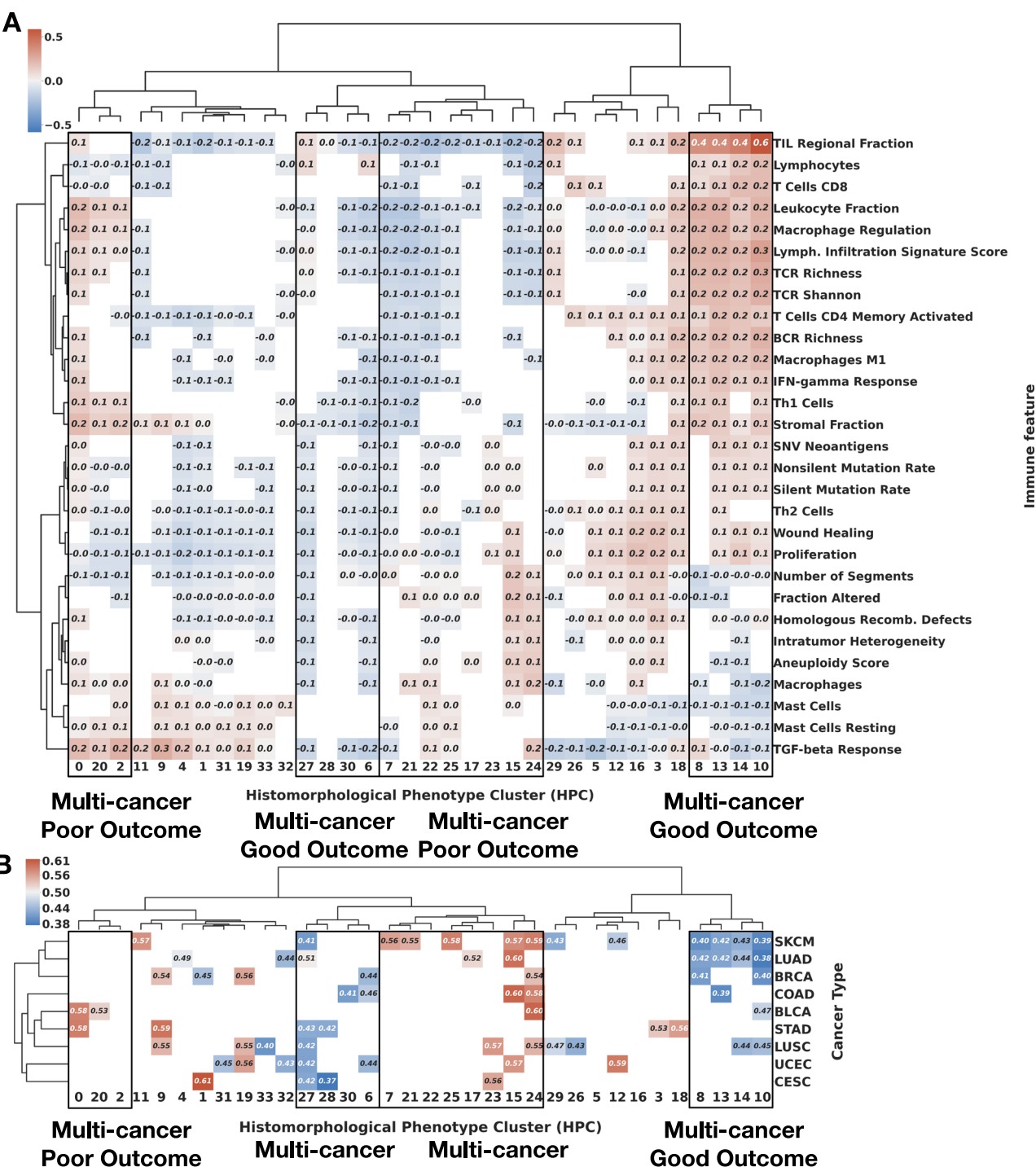

**Fig. 9 | Correlation of multicancer HPCs with immune signatures and survival.**
**A** Full Spearman correlation analysis between HPc and immune features. The correlation values are shown in the graph if their were statistically significant (*p* value < 0.01, two-sided Spearman correlation) and displayed in red for enrichment, and blue for depletion. Two groups of HPCs showing good outcome, and two showing poor outcome are highlighted. Representative tiles from those HPCs are shown in Supplementary Figs. 24, 27. **B** Mean C-index for survival analysis of each HPC and cancer type over a 5-fold cross-validation; values below 0.5 (blue) indicate that higher percentage of the HPC favors longer survival (good outcome), while

those above 0.5 (red) indicate that higher percentage the HPC favors shorter survival (poor outcome). Four hierarchical clusters are almost exclusively associated with C-indexes below 0.5 or, with C-indexes above 0.5 and are highlighted by the multi-cancer poor or good outcome black boxes. Only statistically significant values of log rank test of the high and low-risk groups are displayed (*p* value < 0.05). Supplementary Fig. 23 includes mean and 95% confidence intervals of the C-index values over the 5-fold cross-validation per cancer type. See Methods - Cluster Characterizations for computational details. Source data are provided as a Source Data file.

**Table 1 | Multi-cancer comparison between HPL and relevant methodologies**

| Cancer Type | C-Index HPCs | C-Index AMIL (Porpoise) | C-Index MMF (Porpoise) | C-Index HIPT |
|---|---|---|---|---|
| BLCA | 0.60 | 0.54 | 0.63 | – |
| BRCA | 0.60 | 0.56 | 0.56 | – |
| CESC | 0.63 | – | – | – |
| COAD* | 0.62 | 0.546 | 0.58 | 0.608 |
| LUAD | 0.62 | 0.55 | 0.59 | 0.538 |
| LUSC | 0.57 | 0.56 | 0.53 | – |
| SKCM | 0.61 | 0.61 | 0.61 | – |
| STAD | 0.58 | 0.56 | 0.59 | 0.570 |
| UCEC | 0.57 | 0.64 | 0.65 | – |

AMIL porpoise refers to histology image only attention MIL[30], MMF porpoise refers histology plus RNA seq. model[30], and HIPT refers to the Hierarchical Image Pyramid Transformer architecture[38].* COAD also includes READ in the case of Porpoise and HIPT.

pattern, an approach that is well accepted and corroborated (e.g. the multi-hospital study from *Sun* et al.[67]).

Our methodology's mean concordance-index of 0.74 in recurrence-free survival is in line with the recent IASLC grading recommendation in Moreira et al.[1], which proposed a revised grading system based on expert histopathological assessment of LUAD WSIs but relies on visual inspection from expert pathologists. It is well established by classical pathological studies that the proportions of high-risk growth patterns (solid, micropapillary, and latterly recognised complex glandular patterns such as the cribriform pattern) have predictive value for the risk of lung adenocarcinoma recurrence[68]. HPL similarly finds that poor outcome HPCs are dominated by solid and cribriform patterns as well as necrosis, another recognised correlate of poor outcome[69]. In our subset of 256 WSI for which we have associated International Association for the Study of Lung Cancer (IASLC) grade annotations, we compared the performance of our model versus IASLC grade, achieving c-indexes of 0.72 (95% CI 0.67 − 0.77) versus 0.64 (95% CI 0.58 − 0.71). These results show that HPCs outperform IASLC recommended grading when predicting RFS.

Next, we analyzed the HPCs through three different sources of information immune signatures derived from genomic and transcriptomic profiles, automated cellular annotations from tissue images, and histological subtype annotations from expert pathologists. Along with the survival analysis, these correlations provided a comprehensive overview of the information contained in the HPCs and their predictive potential, showing that tissue patterns that favor better prognostic outcomes suggest an effective immune response[62] and low-grade disease, while HPCs that favor a patient's death or recurrence show high-grade tumor morphology, cellular proliferation, and a lack of immune response. Interestingly, the wound healing process is shown to be positively correlated with HPCs associated with poor prognosis. Mechanisms and pathways associated with wound healing have indeed been pointed as being implicated in cancer proliferation and metastasis[70,71].

We also showed that such a method can be used to answer multi-cancer types questions. First, we assessed its ability to distinguish adenocarcinoma (LUAD) from squamous cell carcinoma (LUSC). The TCGA cohort was used as a training set and an independent cohort from NYU, ($NYU_2$), was used as an independent cohort. The average AUC of 0.99 is even better than results previously obtained by training Inception-V3 on analogous external cohorts[8]. Such results are in line with performance obtained with other recent strategies: AUCs of 0.94-0.96 were obtained using a unified memory mechanism on WSIs rather than tiles[72], AUC of 0.915 using a pre-trained ResNet50 followed by multiple instance learning, or 0.968 using a weak supervision

expectation-maximization framework[73]. In several supervised deep learning approaches, the decision is described as a "black box" and understanding how the classifier makes a decision is not straightforward. In contrast, our methodology provides clear insight into the relevant HPCs and their correlation with the classification task.

The survival prediction in our multicancer survival analysis shows that several HPCs are enriched in universal patterns explainable through correlation with immune feature information. Then, relying exclusively on phenotypes represented by specific HPCs, the performances are on par with those obtained by Chen et al. where the authors used histological and gene expression data[30] or a transformer-based architecture[38] (Table 1, and confidence intervals on our measurements in Supplementary Fig. 23). Clinical prediction of survival is a crucial step impacting disease management[74] and end-of-life decisions[75,76]. Furthermore, the identification of HPCs with multi-cancer outcome implications identifies them as possible sources of target discovery.

Overall, this study shows the potential of HPL, a self-supervised strategy to partition WSIs into meaningful HPCs that can be used to define and quantify morphological phenotypes which recur within and between cases. This process simplifies the task of histological image interpretation, often seen as one of insurmountable complexity, by image conversion to quantitative measures of immediate value in further classification or regression tasks. Our methodology can be applied to any cancer type, offering a powerful discovery approach that can be used to study histomorphological phenotypes across cancer types. We suggest that as these HPCs are refined by repeated observations and refined methodology, many will emerge as stable 'real-world' entities, representing successful recurrent tumor survival strategies worthy of deeper biological study in their own right. Furthermore, the approach has considerable prognostic/biomarker potential. For example, with appropriate clinical trial biopsy images, clusters could be discovered which predict drug response using a model trained on a much larger dataset. The future use of this approach for predictive/prognostic tasks might relieve pathologists of much time-consuming manual interpretation, though reliance on the method will require extensive validation and further characterisation of underlying HPC features. In addition, the clear unassisted emergence of multiple measures of immune infiltration as prognostic features further suggests the clinical adoption of such metrics into therapeutic decisions, whether performed manually or algorithmically, and this is in line with general movements towards integration of TIL counts into pathology reporting[61]. Furthermore, we have shown that the correlation of a cluster with information retrieved from other tools, whether it is sequencing information, features extracted by another segmentation algorithm, or clinical information from pathologists, can generate pathological and biological hypotheses to direct further focused tissue-based research. In the future, we aim to develop this tool, and broaden its application to other tumor types with the aim of discovering recurrent histomorphological phenotypes which underlie the full range of strategies seen in the spectrum of human solid malignancy. Though integrating in our model the few clinical variables available for our dataset did not seem to improve the predictions much Supplementary Figs. 14- 15, current progresses make considering multi-modal data processing (clinical, pathology image, genomics) indeed very exciting as shown by Boehm et al.[77]. We could explore integrating the different modes at various stages of the current framework either at the very end, combining the outcome from the linear or Cox regression with the additional data, or, add it as an additional variable during the regression itself, or, more interestingly, by extending the $z$ vector patient representation generated by Barlow-Twins with additional fields coding for those additional data, which would therefore likely modulate the Leiden clustering representation.

## Methods

Our research complies with all relevant ethical regulations, and the NYU specimens used for this were collected under the NCI/NIH Early Detection Research Network U01CA214195 to Harvey I. Pass MD. NYU slides for this investigation were used according to protocol i8896 "The Lung Cancer Biomarker Center", H. Pass, co-investigator, which was approved by the New York University Langone Health Investigational Review Board on a yearly basis since 2001. All patients signed written informed consent for the use of their tissues, blood, and slides, as well as for the use of corresponding de-identified data, as well as permission to have follow-up by the Principle Investigator. The Race/ Ethnicity of each patient was self-described.

### Datasets

For our lung tumor type prediction task we use The Cancer Genome Atlas (TCGA) and an independent external cohort from New York University (NYU) labeled in this study as $NYU_2$, both formalin-fixed paraffin-embedded (FFPE) and hematoxylin and eosin (H&E) stained whole slide images (WSIs). The TCGA cohort is composed of 1021 WSIs, 513 of adenocarcinoma (LUAD) and 508 WSIs of squamous cell carcinoma (LUSC). $NYU_2$ is composed of 138 WSIs, 72 of LUAD and 66 of LUSC and comparable to the one used in *Coudray et al.*[8]. Whenever necessary, the pathologist' diagnosis was supplemented by immunohistochemical stains (TTF-1 and p40 for LUAD and LUSC respectively), which results in a clean dataset with reliable labels.

The adenocarcinoma overall survival analysis used the TCGA LUAD cohort and an additional cohort from NYU[1] labeled in this study as $NYU_1$. The TCGA cohort is composed by 442 patients with a censorship rate of 0.64 and all stages (Stage 1-4), $NYU_1$ is composed of 276 patients with a censorship rate of 0.80 and only Stage 1 cases. Further details on both cohorts, such as Kaplan-Meier curves and a distribution comparison between follow-up times can be found in the Supplementary Fig. 28.

The adenocarcinoma recurrence-free survival analysis focuses on recurrence types of systemic and loco-regional from $NYU_1$. It is composed of three types: systemic, loco-regional, and new primary. We censored new primary cases as we focus on systemic and loco-regional. The motivation behind only using the NYU cohort is based on the more detailed recurrence information and longer follow-up times. This dataset is composed of the same 276 patients from $NYU_1$ used in the overall survival analysis, the censorship rate for recurrence is 0.82. In addition, 70 NYU WSIs have nonexhaustive manual annotations from pathologists on histological subtypes (solid, papillary, micro-papillary, acinar, and lepidic) allowing individual tile labeling. Further details on the recurrence-free survival cohort can be found in the Supplementary Fig. 28.

Our multi-cancer analysis used 10 cancer types from The Cancer Genome Atlas (TCGA). The complete cohort is composed of 279 patients of bladder urothelial carcinoma (BLCA), 364 patients of breast invasive carcinoma (BRCA), 187 patients of cervical squamous cell carcinoma and endocervical adenocarcinoma (CESC), 369 patients of colon adenocarcinoma (COAD), 366 patients of lung adenocarcinoma (LUAD), 367 patients of lung squamous cell carcinoma (LUSC), 247 patients of prostate adenocarcinoma (PRAD), 363 patients of skin cutaneous melanoma (SKCM), 278 patients of stomach adenocarcinoma (STAD), and 395 patients of uterine corpus endometrial carcinoma (UCEC). Cancer types and number of patients are an intersection between available TCGA cancer types and immune signature information[54]; we also selected cancer types with at least 150 annotated samples.

The immune landscape features used in the results section are derived from a immune signature analysis of 33 cancer types published by *Thorsson et al.*[54]. These immune signatures were derived from bulk RNASeq data from TCGA.

### Whole slide image pre-processing

We used the publicly available pipeline DeepPATH[8] to divide each WSI into non-overlapping tiles of 224 × 224 at 5X magnification (with tiles re-sampled to ensure 2.016 µm per pixel) and at 20X magnification (with tiles re-sampled to ensure 0.504 µm per pixel), filtered out images with less than 60% tissue in total area (considering background pixels those which average grey-level is above 230), and applied stain normalization using Reinhard's method[42], which aims to match the mean and standard deviation of an image to a target. Instead of normalizing to a given image reference, the average values of 100 random tiles from the TCGA dataset were taken as the target value.

### Self-supervised learning

Self-Supervised learning aims to create useful representations of data based on its features, which unlike supervised methods, does so without relying on expensive human annotations. These methods use a pretext task as a mean to capture relevant feature information into each instance representation. Subsequently, these representations can be used in downstream tasks.

A common setup in self-supervised learning is the use of a data augmentation pipeline $T$ in order to create invariant representations against image distortions[78] and a backbone network $f_\theta$ to project the images into a latent space $z$, where the different approaches define the pretext task goal.

Barlow Twins[44] aims to create representations that are invariant to distortions while minimizing the redundant information between feature dimensions in the representation. It uses two different transformations to distort each image in the batch $\{\tilde{x}_i = t(x_i), \tilde{x}'_i = t'(x_i); t \sim T, t' \sim T \text{ and} [x_n]_{n=1}^N\}$ and runs the images through the backbone network $f_\theta$ to obtain latent representations $z \in R^D / D = 128; z_i = f_\theta(\tilde{x}_i), z'_i = f_\theta(\tilde{x}'_i)$. Right after, the representations are batch normalized for each transformation set $Z \in R^{NxD}; \{Z, Z'\}$ and calculates $C \in [-1, 1]^{DxD}$ as the cross-correlation matrix computed between $\{Z, Z'\}$ along the batch dimension.

The loss is defined by two terms, the invariance term that attempts to create representations invariant to distortions and the redundancy reduction term where off-diagonal features of the correlation matrix are minimized to prevent any redundant information between them (Equation (2)). The parameter $\lambda$ is used as a weighting factor to balance the difference in number between the diagonal and off-diagonal terms:

$$\mathscr{L} = \underbrace{\sum_i^D (1 - C_{ii})^2}_{\text{invariance}} + \lambda \underbrace{\sum_i^D \sum_{j \neq i}^D C_{ij}^2}_{\text{redundancy reduction}}; \quad C_{ij} = \frac{\sum_{n=1}^N z_i \, z'_j}{\sqrt{\sum_{n=1}^N (z_i)^2} \sqrt{\sum_{n=1}^N (z_j)^2}}$$

(2)

It is relevant to mention how Barlow Twins avoids trivial solutions to the pretext task, also known as collapse in self-supervised learning, where representations for different images hold constant values. This method avoids collapse via redundancy reduction, forcing decorrelation between features of the latent representations, and by batch normalizing the batch vector representations before computing the cross-correlation matrix. It has be shown not to be dependent on batch sizes as contrastive methods[45,79].

The backbone network $f_\theta$ is a convolutional neural network (CNN) used as a feature extractor of the tissue images. In our work, we used a CNN composed of several ResNet layers[80] and a self-attention layer[81,82]. This architecture is similar to the one used in *Quiros et al.*[36], where it was shown to be an effective feature extractor.

More specifically, the backbone CNN network is composed of 19 layers (conventional convolutional and ResNet) and an additional self-attention layer at the feature maps 56 × 56x32. The kernels sizes range from 7 × 7, 5 × 5, and 4 × 4 in comparison to conventional 3 × 3 (e.g.

ResNet-50); we introduce these changes to capture a larger receptive field of interaction. The number of trainable parameters of the entire network is $21,197,785$. In addition, we use spectral normalization, a normalization technique which has been shown to effectively limit the size of gradient updates and stabilize training[83]. We include a detailed description of the CNN layers in the Supplementary Fig. 29.

We trained the self-supervised model using 250K tile images for 60 epochs and a batch size of 64. We used a single GPU (NVIDIA Titan Xp 12GB/NVIDIA RTX 24GB) with training times ranging between 48 to 72 hours. Our approach allows training the self-supervised model in a reasonable amount of time without the need for a large GPU cluster. Once the backbone network $f_\theta$ was trained, it was frozen and only used to project tissue tiles into representations.

To ensure training on a subset of tiles is not affecting the performance, we also ran the training and HPL pipeline using the full tiles from the training set, using a 5-fold cross-validation approach and the training/validation split similar to one used in previously published articles[30,38], and detailed splits are available for download on our github page https://github.com/AdalbertoCq/Histomorphological-Phenotype-Learning/tree/master/utilities/files/LUAD.

In addition, we compared HPL performance replacing Barlow Twins with DINO, another relevant self-supervised learning method. Supplementary Fig. 13 shows a comparison on LUAD survival performance between these two methods. Overall, the performance of both self-supervised methods is comparable. However, DINO concordance index fluctuates more with respect to the Leiden resolution parameter (resulting in more/less cluster) while Barlow Twins remains rather stable at different resolutions.

## Clustering representations

Leiden community detection[48] has been shown to be a successful algorithm for discovering well-connected communities in graph structures. Derived as an improvement from Louvain algorithm[84], it uses a heuristic method to find partitions in graphs.

The algorithm starts by assigning each node in the graph to a different community, then it iterates through the steps described below until there are no further changes in the network. First, for each node the algorithm will evaluate the gain of modularity when the node is moved from the current community to a neighboring one; if there's a positive gain, the node is kept in the new community. After an initial pass, this process is repeated for nodes that changed community until it reaches a local maximum, finding a partition of the graph. Secondly, the previous partition is refined by possibly further splitting some of the previously defined communities. This is done by randomly merging a node to a community if it increases modularity, which allows to further explore the partition space and avoids badly connected communities, a shortcoming from Louvain algorithm[48]. Finally, nodes in each community are aggregated, and communities are treated as nodes in the next iteration.

Modularity is defined by Equation (3), where $e_c$ is the number of edges in a community c, $\frac{K_c^2}{2m}$ is the expected number of edges where $K_c$ is the sum of the degrees of the nodes in a community c, and m is the total number of edges in the network. $\gamma$ is the resolution parameter where higher values lead to more communities and lower values to fewer communities:

$$\mathcal{H} = \frac{1}{2m}\sum_c \left(e_c - \gamma\frac{K_c^2}{2m}\right) \qquad (3)$$

In order to find Histomorphological Phenotype Clusters (HPCs) from self-supervised tissue representations, we created a graph by using K nearest neighbors ($K = 250$) over 200,000 randomly sampled tiles from the training set, and used Leiden community detection to define clusters (note: for studies where the heterogeneity and imbalance of the training dataset result in poor performances on the cross-

validation and external cohort, users of our pipeline may want to consider using more tiles and balance the training datasets using techniques commonly used in the field such as oversampling the minority dataset, or data augmentation of one of the sets for example). Subsequently, we assigned clusters to vector representations of additional sets by using again K nearest neighbors between vectors of the training set and each of the vectors of the additional set.

We performed the clustering process an initial time in an effort to identify and remove representations of background and artefact tiles. Once these representations were removed, we run the clustering process an additional time so these noninformative tile representations do not interfere with the number of tissue communities found from the Leiden algorithm.

Finally, we propose the method below to set $\gamma$, and therefore select the number of HPCs. In order to balance how closely HPCs are packed together and their ability to generalize across cohorts from different institutions, we framed a trade off between the HPC compactness ($C_{\gamma_i}$ - Equation (4)) and average presence of institutions per HPC ($AIP_{\gamma_i}$ - Equation (5)). This Leiden resolution selection method is specific to the question we want to address here, using the TCGA multi-institution dataset as training, and for which we want to want to identify features which are as little affected as possible by the institution it comes from. Different objectives may require selecting different threshold selection strategies.

For each $\gamma_i$ resolution value (set of HPCs), we measure compactness $C_{\gamma_i}$ as the average Euclidean distance $d(.,.)$ between the centroid $\mu_n$ of an HPC n, and each tile vector representation $z_i$ belonging that HPC n. This is shown in Equation (4), where K describes the total number of tile vector representations and M is the number of tile vector representation belonging to an HPC.

$$C_{\gamma_i} = \frac{1}{K}\sum_{k=1}^{K} d(z_k,\mu_J) \text{ where } \mu_n = \frac{1}{M}\sum_{p=1}^{M} z_p \qquad (4)$$

Also, for each $\gamma_i$ resolution value, we measure the average presence of institutions per HPC $AIP_{\gamma_i}$ as the weighted sum of the relative size of the HPC and the percentage of institutions present in the HPC. We use the relative size of the HPC as a weighting factor in order to penalize HPC bias accordingly to its impact over the entire partition: larger HPCs that are highly biased will have a larger impact on any downstream task rather than smaller ones. In Equation (5), we define N as the number of HPCs for a given $\gamma_i$, $t_n$ as the number of institutions present in the HPC n, and T as the total number of TCGA institutions. We also define $s_n$ as the relative size of HPC n, computed as the number of tile representations belonging to that HPC divided by the total number of tiles K.

$$AIP_{\gamma_i} = \frac{1}{N}\sum_{n=1}^{N}\left[s_n * \frac{t_n}{T}\right]; \; AIP_{\gamma_i} \in (0,1) \qquad (5)$$

In addition, to achieve the trade-off mentioned earlier, we invert the compactness $C_{\gamma_i}$ and weight it by the average institution presence $AIP_{\gamma_i}$. In this setting, a more compact set of HPCs will get closer values to the maximum value $max(C_\gamma)$ and they will be penalized by their ability on generalizing across institutions $AIP_{\gamma_i}$ (Equation (6)):

$$Score_{\gamma_i} = [max(C_\gamma) - C_{\gamma_i}] * AIP_{\gamma_i} \qquad (6)$$

Finally, we selected the optimal Leiden resolution based on where the score function shows a larger trend change and begins saturation (elbow method or knee of a curve). In Supplementary Fig. 1 we show, for each of the three studies presented in this paper, the final score plot used to identify the optimal Leiden resolution, as well as the intermediate plots leading to that curve.

## Whole Slide Image and patient vector representations

We defined whole slide image and patient (one or more WSIs) representations as a vector with dimensionality equal to the number of Leiden clusters $C$ (i.e. Histomorphological Phenotype Clusters (HPCs)), where each dimension describes the percentage contribution of an HPC to the total tissue area (Equation (7)):

$$w = \{w_0, w_1, ..., w_{C-1}\} \text{ where } \{w_i \in [0, 1] / \sum_{i=0}^{C-1} w_i = 1\} \tag{7}$$

In order to use these WSI vector representations in linear models we perform the center log-ratio transformation (Equation (8)). Models such as logistic regression or Cox proportional-hazards require independence between the covariates. The original definition of the WSI vector representations is an example of compositional data which violates this assumption ($\sum_{i=0}^{C-1} w_i = 1$), the center log-ratio transformation breaks the co-dependency between variables facilitating the use of these models[85]. In addition, we use multiplicative replacement[86] to avoid zero values before applying center log-ratio transformation.

$$clr(w) = \left\{ log \frac{w_0}{g(w)}, log \frac{w_1}{g(w)}, ..., log \frac{w_{C-1}}{g(w)} \right\} \text{ where } g(w) = \left( \prod_{i=0}^{C-1} w_i \right)^{1/C} \tag{8}$$

## Evaluation and coding

To train the self-supervised model, we randomly selected 678 slides out of all 1021 LUAD and LUSC TCGA WSIs, leaving the NYU cohorts unseen. In addition, we subsampled 250K of 583K tiles from those selected WSIs. Once the self-supervised model was trained, it was frozen and used to project tissue tiles into representations. To ensure sub-sampling was not affecting the performance, the LUAD survival analysis task was also done after training using the full nonsubsampled training cohort (and using a 5-fold cross-correlation split similar to the one used in Chen et al.[30,38]). We tracked the train and validation loss of the model during training to ensure no overfitting. We find that both losses converge concurrently without overfitting (Supplementary Fig. 30).

For our lung adenocarcinoma and squamous cell carcinoma classification analysis we perform the following procedure. We set up a 5-fold cross-validation, and to further prevent site-specific bias[43], we performed it in a way to obtain non-overlapping institutions between TCGA training, validation, and test sets. For each fold, we defined Leiden clusters based on representations of the TCGA training set and assign clusters (i.e. HPCs) to TCGA validation and test representations based on those found in the training set. The cohort from NYU was used as an additional external test set to which those HPCs were assigned as well. Subsequently, we fit a logistic regression using WSI vector representations from the training set and evaluated performance on TCGA validation and test sets, and NYU cohort. The motivation behind this procedure was to confirm that the methodology is robust against different imaging processes across institutions, maintaining consistent performance. These results can be found in Supplementary Fig. 31. Once we established that HPCs on different training sets provide similar performance, we locked down a particular cluster configuration and used it to evaluate the performance of the logistic regression across the 5-fold cross-validation. We used this procedure so clusters are common across folds, and we can evaluate their impact to predict lung adenocarcinoma or squamous cell carcinoma.

We used SHAP (SHapley Additive exPlanations) to evaluate the impact of an HPC into the log odds ratio for each patient. We calculated SHAP values across each test set of the 5-fold cross-validation (Supplementary Fig. 16D). This approach allows us to check the consistency of the relationship between the HPC contribution value and its effect

on the log odds ratio across folds; the HPC value (intensity of red/blue) and its relation to the model output should be similar across folds. We used this method instead of a Forest plot for each coefficient since our logistic regression is L1-norm penalized, and confidence intervals of coefficients can be optimistic for penalized regressions[87–89].

In addition, we do not evaluate WSIs with less than 100 tiles; we make this choice in order to ensure that we get enough HPC assignations from the tissue tiles. This operation results in not evaluating 3.8% (39/926) of the WSIs.

We follow a similar process for our LUAD overall survival analysis, performing a separate independent clustering for the sake of consistency. We setup a 5-fold cross-validation using the TCGA dataset, dividing it into a training and test set, and kept NYU as an additional independent test set. In this case, we did not consider a TCGA validation set or non-overlapping institutions due to the limited number of samples and in an effort to have balanced folds. For each fold, we defined HPCs based on representations of the TCGA training set and assigned HPCs to representations of TCGA test and NYU based on those found in the training set. Afterwards, we used Cox proportional hazards regression to model overall survival, we fit the mode using WSI vector representations from the training set and evaluated the performance on TCGA test and NYU sets. As in the case of the lung type classification, the motivation for this approach was to show that the methodology is robust even when clustering is done across different training sets. These results can be found in Supplementary Fig. 31.

Then, we locked down a particular cluster configuration to find which HPCs impact the prediction of overall survival in the Cox model. We used SHAP to evaluate the impact of an HPC into the log hazard ratio for each patient. We calculated SHAP values across each test set of the 5-fold cross-validation (Fig. 6). We used this method instead of a Forest plot since our Cox proportional hazards model is L2-norm penalized, and confidence intervals of coefficients can be optimistic for penalized regressions.

In addition, we do not evaluate patients with less than 100 tiles; we make this choice in order to ensure that we get enough HPC assignations from the tissue tiles. This operation results in not evaluating 6.5% (28/442) of TCGA and 2.9% (8/276) of $NYU_1$ patients.

On our LUAD recurrence-free survival analysis, we used a 5-fold cross-validation over the NYU cohort. In this case, we use the same HPCs defined by the TCGA training sets of overall survival.

Finally, we used the same cluster configuration selected in overall survival. This allows us to relate the impact of HPCs in recurrence free and overall survival. Once again, we used SHAP to evaluate the impact of an HPC in into the log hazard ratio for each patient; calculating values across each test set of the 5-fold cross-validation (Fig. 6C).

High and low-risk groups were defined by the median of the hazard predictions of the training set. This median value was later used to divide risk groups for the test and additional independent sets. We used log rank test to measure statistical significance between risk groups and use a $p$ value threshold of 0.05.

For our multi-cancer overall survival analysis, we directly used the patient HPC contribution to model risk and calculate concordance index (c-index). This approach does not require any kind of parameter fitting, and the calculated c-index can be interpreted as an HPC correlating with a death event (c-index > 0.5) or survival (c-index < 0.5). For each cancer type, we performed a 5-fold cross-validation by dividing the complete set of patients into 5 different sets of equal or similar size and evaluating the c-index in each of them. In Supplementary Fig. 23, we show the mean and 95% confidence interval for the c-index values across the 5-fold cross-validations per cancer type. In addition, we only display c-index values where the log-rank test of the high and low-risk groups is significant ($p$ value < 0.05).

The code is written is Python, and deep learning models are implemented in TensorFlow. We used Leiden[48] and PAGA[90] algorithm implementations from ScanPy[91], Cox proportional hazards and Kaplen-

Meier models from Lifelines[92], statistical tests (Fisher's exact, hypergeometrical, and Kolmogorov-Smirnov) from SciPy[93], and logistic regression implementation from statsmodels[94].

## Cluster characterizations

Correlation analysis between HPCs (centered log-ratio on the percentage of HPC per patient) and immune landscape features (immune feature expression)was done per patient through Spearman's rank correlation. First, patient vector representations were constructed as a vector with dimensionality equal to the number of HPCs, where each dimension describes the percentage contribution of an HPC to the total tissue area. Second, the patient vector representations were normalized by the multiplicative replacement (delta of 1/$(numberHPC)^2$, to avoid zero values) and center log-ratio transformation. This results in each patient having a continuous value for each immune signature expression (e.g. TIL Regional Fraction raging from 0 to 35). For each HPC and immune signature pair, we performed Spearman correlation over all patients. The significance threshold was set to be 0.01 of the adjusted $p$ value and $p$ values were adjusted for multiple comparisons through the Benjamini/Hochberg method for false discovery rate[95]. Results are shown in Fig. 7A.

Multi-cancer HPCs correlations with immune landscape features were reported by calculating the mean value of correlation per cancer type. P-values across cancer types were combined through Fisher's combined probability test and the significance threshold was set to be 0.01. Results are shown in Fig. 9A.

To examine the enrichment and depletion of cell types in tissue tiles of each HPC, we used Hover-Net[55] to estimate counts of neoplastic, connective, inflammatory, and dead cells in each tile of the lung adenocarcinoma $NYU_1$ cohort. These annotations allow us to measure the distribution of cell type counts per tile for each HPC and the entire population of tiles. We measured the over and under-representation of each cell type in an HPC by measuring the distribution shift between the entire population of tiles and tiles assigned to specific clusters. We used the Two-sample Kolmogorov-Smirnov (K-S) test to account for this distribution shift and used the K-S statistic $D_{n,m}$ to quantify for over-representation and under-representations, assigning $+D_{n,m}$ if there is over-representation and $-D_{n,m}$ if there is under-representation. The Two-sample Kolmogorov-Smirnov test uses the statistic $D_{n,m}$ to quantify the distance between empirical cumulative distributions. The statistic is defined as $D_{n,m} = sup_x|F_{1,n}(x) - F_{2,m}|$ where $F_{1,n}(x), F_{2,m}$ are the empirical cumulative distribution functions of the first and second samples, $n$ is the sample size for the $F_1$, $m$ is the sample size for the $F_2$, and $x$ the support for both distributions. The significant threshold is set to be 0.01 of the adjusted p-value and p-values were adjusted for multiple comparisons through the Benjamini/Hochberg method for false discovery rate[95]. Results are shown in Fig. 7B. An example for over and under-representation and how it translates to the statistic $D_{n,m}$ can be found in the Supplementary Fig. 32.

Additionally, we measured the enrichment of lung adenocarcinoma subtypes such as solid, acinar, papillary, micropapillary, and lepidic for each HPC (Fig. 7A–C). The lung adenocarcinoma $NYU_1$ cohort contains manual annotations from **3** pathologists on these histological subtypes (done on a per slide level using ImageScope software; Aperio Technologies, Vista, CA, USA), and we translated the manual region annotations into individual tile annotations (each tile was assigned the label of the region drown by the pathologist if that region had more than 50% overlap with the tile). Subtype enrichment is measured by using the hypergeometric test comparing each HPC to the entire population of annotated tiles.

The hypergeometric test uses the hypergeometric distribution in order to measure the statistical significance of seeing $k$ successes in $n$ draws from a population of size $N$ with $K$ successes, without replacement. In our case, we specify $k$ as the number of tiles with of a particular histological subtype given as the number of total tiles in the HPC $n$, $K$ is the number of tiles with the same particular histological subtype in the entire population of annotated tiles of size $N$. We measure enrichment by using a *fold* metric defined as $k/\mathbb{E}[x]$ where $\mathbb{E}[x]$ is the expectation of the hypergeometric distribution given $n$, $K$, $N$. Enrichment for a particular histological subtype shows as *fold* > 1 and depletion as *fold* < 1, we use a significance threshold of 0.01 for the adjusted p-value. P-values were adjusted for multiple comparisons through the Benjamini/Hochberg method for false discovery rate[95]. Examples of how we used hypergeometric test for enrichment and depletion can be found in the Supplementary Fig. 33.

## Cluster histological assessment

The HPCs defined for LUAD were examined independently by three subspecialty diagnostic histopathologists (J.L.Q., N.N., D.M.).

For each HPC 100 randomly selected tile images per cluster were assessed. Each tile had a resolution of 224 × 224 pixels at 5X magnification, covering approximately 452 μm² of tissue (2.016 μm per pixel), based on a previous demonstration that such resolution and field of view are sufficient for many lung classification purposes[8]. Each HPC was morphologically described and classified using a structured data collection tool. Each cluster was classified as being either predominantly made up of tumor-bearing tiles or not. Depending on this classification, pathologists were asked to list the predominant and second-most predominant features (tumor growth pattern or non-tumor components, respectively) from a set of options. In either case, pathologists provided information on lymphocytic infiltration (ie the amount of infiltrating TILs) and necrosis, and were encouraged to make free text comments. These expert annotations were converted into consensus measures, including an agreement score. The agreement score gives the number of pathologists agreeing on the most abundant pattern or feature (1–3). Values less than 3 often reflect situations where two or more appearances are present at near-equal levels, although there is also known considerable variance in such histopathological judgements. For each cluster, a short summary title was created, which summarises the histopathological opinions and describes the overall dominant appearance of the HPC. Results shown in Figs. 2D, 3 and 4.

## Reporting summary

Further information on research design is available in the Nature Portfolio Reporting Summary linked to this article.

# Data availability

The Cancer Genome Atlas (TCGA) whole slide images and corresponding labels for the 10 cancer types (accessing IDs TCGA-LUAD, TCGA-LUSC, TCGA-BLCA, TCGA-BRCA, TCGA-CESC, TCGA-COAD, TCGA-PRAD, TCGA-SKCM, TCGA-STAD) are available at the Genomic Data Commons portal (https://gdc.cancer.gov/). This data is publicly available without restriction, authentication or authorization necessary. The immune landscape signatures for the TCGA samples are available in *Thorsson et al.*[54]. Due to privacy, ethical considerations, and in accordance with the institutional policies, requests for whole slide images and corresponding labels in the additional New York University cohorts data may be addressed to the corresponding author, and a data transfer agreement between institutions will need to be signed - length and conditions for access will be defined by the parties following the procedure described in https://hslguides.med.nyu.edu/datasharing. The data generated in this study (pre-trained LUAD/LUSC model checkpoints, multi-cancer model checkpoints; tile vector representations for LUAD/LUSC before and after artefact removal; tile vector representation for the multi-cancer model; HPC configurations used in the publication for background and artefact removal, for LUAD/LUSC type classification, for LUAD survival and for multicancer analysis; whole slide image and patient vector representations for the

lung subtype classification, the LUAD survival and the multicancer study; jupyter notebook to generate figures and results presented here) are available for download from the github page https://github.com/AdalbertoCq/Histomorphological-Phenotype-Learning. Source data are provided with this paper.

## Code availability

Code, pre-trained models, and demonstrations can be found at: https://github.com/AdalbertoCq/Histomorphological-Phenotype-Learningand on Zenodo[96].

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

## Acknowledgements

The authors thank the team of the Center of Biospecimen Research and Development from the New York University for their help in digital pathology and histology for this project. The Center of Biospecimen Research and Development, RRID:SCR_018304, is partially supported by the Cancer Center Support Grant P30CA016087 at the Laura and Isaac Perlmutter Cancer Center. We would like to thank the Applied Bioinformatics Laboratories (ABL) for providing bioinformatics support and helping with the analysis and interpretation of the data. GTC and ABL are shared resources partially supported by the Cancer Center Support Grant P30CA016087 at the Laura and Isaac Perlmutter Cancer Center. This work has used computing resources at the NYU School of Medicine High Performance Computing (HPC) Facility. K.Y. acknowledges support from EP/R018634/1 and BB/V016067/1. J.L.Q. is supported by the Mazumdar-Shaw Molecular Pathology Chair endowment at the University of Glasgow. A.C.Q. is supported by a scholarship from School of Computing Science, University of Glasgow. K.Y. and A.C.Q. acknowledge support from the Openshift GPU cluster management team. B.L. is supported by the Swedish Research Council (BL, 2019-06360). A.T. acknowledges support from NCI/NIH Cancer Center Support Grant P30CA016087.

## Author contributions

A.C.Q. and N.C. designed and executed the experiments, prepared the manuscript, and wrote the Python package. A.Y. and X.Y. partially supported a subset of the experiments. H.L. helped with experiments at 20x. H.P., A.L.M. and L.C. put together the NYU cohort dataset and associated information. A.K. and N.N. provided slide annotations of LUAD histological subtypes. J.L.Q. and D.A.M. provided the histological assessment of HPCs for the LUAD vs LUSC and LUAD overall and recurrence free survival studies. B.L. provided constructive feedback during the development of the method and redaction of the manuscript. C.Y.P. provided supervision for some of the annotations. J.L.Q., A.T., and K.Y. designed the experiments, edited the manuscript, and jointly supervised the research.

## Competing interests

The authors declare the following competing interests: A.T. is a cofounder of Imagenomix; N.C. is a scientific advisor for Imagenomix. The other authors declare that they have no competing interests.

## Ethics

NYU specimens used for this were collected under the NCI/NIH Early Detection Research Network U01CA214195 to Harvey I. Pass MD. NYU slides for this investigation were used according to protocol i8896 "The Lung Cancer Biomarker Center", H. Pass, co-investigator, which was approved by the New York University Langone Health Investigational Review Board on a yearly basis since 2001. All patients signed written informed consent for the use of their tissues, blood, and slides as well as for the use of corresponding de-identified data, as well as permission to have follow-up by the Principle Investigator. The Race/Ethnicity of each patient was self described.

## Additional information

[1]School of Computing Science, University of Glasgow, Glasgow, Scotland, UK. [2]School of Cancer Sciences, University of Glasgow, Glasgow, Scotland, UK. [3]Applied Bioinformatics Laboratories, NYU Grossman School of Medicine, New York, NY, USA. [4]Department of Cell Biology, NYU Grossman School of Medicine, New York, NY, USA. [5]Department of Medicine, Division of Precision Medicine, NYU Grossman School of Medicine, New York, USA. [6]Department of Pathology, NYU Grossman School of Medicine, New York, NY, USA. [7]Department of Medical Epidemiology and Biostatistics, Karolinska Institutet, Soln, Sweden. [8]Department of Cellular Pathology, University College London Hospital, London, UK. [9]Cancer Research UK Lung Cancer Centre of Excellence, University College London Cancer Institute, London, UK. [10]Department of Cardiothoracic Surgery, NYU Grossman School of Medicine, New York, NY, USA. [11]Cancer Research UK Scotland Institute, Glasgow, Scotland, UK. [12]Queen Elizabeth University Hospital, Greater Glasgow and Clyde NHS Trust, Glasgow, Scotland, UK. [13]These authors contributed equally: Adalberto Claudio Quiros, Nicolas Coudray. [14]These authors jointly supervised this work: John Le Quesne, Aristotelis Tsirigos, Ke Yuan. ✉e-mail: John.LeQuesne@glasgow.ac.uk; Aristotelis.Tsirigos@nyulangone.org; Ke.Yuan@glasgow.ac.uk

