## [Peer Review File · Nature Communications]

Editorial Note: This manuscript has been previously reviewed at another journal that is not operating a transparent peer review scheme. This document only contains reviewer comments and rebuttal letters for versions considered at Nature Communications. Mentions of the other journal have been redacted.

REVIEWER COMMENTS

Reviewer #1 (Remarks to the Author):

EDITORIAL NOTE: Reviewer #4 assessed your response to Reviewer #1, and commented that most concerns were addressed. In relation to this report, they recommend that you only use the "self-supervised" terminology throughout your manuscript and avoid "unsupervised".

Reviewer #2 (Remarks to the Author):

EDITORIAL NOTE: Reviewer #3 assessed your response to Reviewer #2. They consider that the following original concerns were not fully addressed: #4 regarding the usefulness of a C-index of 0.74; #6 regarding potential inflation of the quoted performance and whether the task being solved could be trivial or already solved; #7 regarding a lack of novel findings.

Reviewer #3 (Remarks to the Author):

I believe that the authors have addressed most of my points first raised in review at [Redacted]. However and critically, whilst I think the use of self-supervised learning and clustering / community detection is a hot topic and certainly hold promise in the field of histopathology, the applications have to demonstrate utility to the field (e.g. oncology here) but those presented in the manuscript are not convincing, non-novel, and unfortunately make up the majority of the manuscript.

My concerns are shared with 2 other reviewers and centre around novelty of HPL applications. For example, finding that TIL macrophages, proliferation or TGF- β response associates with patient survival is a known finding. Having a C-index performance of approximately 0.65 is not clinically useful and is below

other supervised approaches available. Whilst it's reassuring to have positive controls (i.e. known associations recapitulated) we also would like to see novel inferences previously unexplored.

One important point I made in my review which was ignored, was that Barlow-twins as a self-supervised model, has been shown to be inferior to other self-supervised methods, namely DINO. For example, the latest Top-1 accuracy for self-supervised methods is DINOv2 (#1 - 86.7%) versus Barlow Twins (#73 - 73.2%). Giving the authors the benefit of the doubt as DINOv2 is a recent development, we can compare Barlow Twins to DINOv1 (#16 - 80.3%). Both Barlow Twins and DINO were published in the same year (2021), so it is unclear to me why the authors would choose an inferior method.

In summary, whilst I very much like the self-supervised approach + community detection, I, along with other reviewers, do not think its downstream applications in oncology presented here are sufficiently novel to warrant publication.

Reviewer #4 (Remarks to the Author):

Since this is the second round of review, I am only focussing on the points that were raised and my thoughts on the author response corresponding to those. I am not fully satisfied with the author response to points numbered 2 and 3 raised in my last report as follows.

Regarding point 2 in previous comments, I would still encourage the authors to do a conformal analysis, it is quite a generalised approach and can be easily incorporated into the current setup. That would be a good way to perform the uncertainty quantification. <https://arxiv.org/abs/2107.07511>

Regarding point 3 in previous comments, you have used TCGA data, which often contains other data modalities (like radiology, genomics) and also patient metadata table (age, sex, etc), where there any for yours? In that case, could you comment or show some analysis on the multi-modal question raised specific to your data instead of a general comment?

Reviewer #1 (Remarks to the Author):

EDITORIAL NOTE: Reviewer #4 assessed your response to Reviewer #1, and commented that most concerns were addressed. In relation to this report, they recommend that you only use the "self-supervised" terminology throughout your manuscript and avoid "unsupervised".

We thank the reviewers for this note on consistency. We have modified the text accordingly.

Reviewer #2 (Remarks to the Author):

EDITORIAL NOTE: Reviewer #3 assessed your response to Reviewer #2. They consider that the following original concerns were not fully addressed:

#4 regarding the usefulness of a C-index of 0.74;

We have done an additional analysis to compare the performance of our model against the current International Association for the Study of Lung Cancer (IASLC) recommendation which proposes grading as the new standard approach to assessing patient risk in early-stage lung cancer (Moreira et al, J. Thoracic Oncology, 2020). In this analysis we compare the performance of morphological grade and the HPC model at predicting recurrence-free survival (RFS). Grading was provided by Dr. Andre Moreira, the first author of the IASLC recommendation study referenced above.

Of note, in our original RFS analysis we used all 276 patients of the NYU₁ cohort, however only a subset of 256 have an associated grade annotation (Moreira et al. 2020), therefore we only used the subset of 256 patients in the following results.

The HPC model achieves a mean C-index of 0.72 while only morphological grade reaches 0.64. We can conclude that HPC information completely outperforms morphological grade alone when predicting RFS.

Recurrence Free Survival	HPCs	Morphological Grade
C-Index Mean (95% CI)	0.72 (0.67-0.77)	0.64 (0.58- 0.71)

Original comment: "Going back to the big picture/claim on biomarkers, a C-index of 0.74 is good, but not outstanding and not yet shown to out-compete biomarkers currently available for predicting the recurrence of LUAD. "

By inclusion of the additional multivariate model outlined above, we now demonstrate the superiority of our method, which to our knowledge outperforms all other published methods which predict recurrence-free survival in LUAD from histopathological appearances alone. We previously commented on the performance of our model against non-image-based methods:

“The C-index of 0.74 is far from perfect, but it is at least equal to the best-published performance yet obtained in predicting outcomes from H&E images alone from histological grading (Moreira et al, 2020) or morphological features from histopathological images (Wang et al. 2017, Yu et al. 2016). It may be near the theoretical ceiling of available information, given the amount of prognostic clinical information which is unavailable in an image. Other predictions include genes for pro- and anti-inflammatory cytokines have been shown to identify the subset of stage I patients with poor prognosis (Seike et al. 2007). A study from 17 randomized clinical trials showed that Biomarker status was the strongest predictor of PFS with c-index of 0.73 (Siah et al. 2019). But those are not image-based. “

#6 regarding potential inflation of the quoted performance and whether the task being solved could be trivial or already solved;

Regarding potential inflation of the quoted performance and whether the task being solved could be trivial or already solved, we already agreed with Reviewer #2 that the problem is indeed already solved, and we followed their advice to shorten that section.

Original comment: “6. In classification of LUAD and LUSC, a task that is already well achieved in current literature, the authors showed that HPL achieved an average TCGA test set AUC of 0.930 and an average of 0.990 with a very tight 95%CI on the independent test cohort from NYU. This result is too good to be true to the reviewer. One possible reason for an independent validation result to be better than the internal training data is the task is too easy for the validation set, hence can be achieved without the need for new methods like HPL. **I would shorten this whole section and focus on using this as a case study to validate HPL, and to compare with other imaging-based methods, which are currently missing.**”

#7 regarding a lack of novel findings.

Original comment: “Figures 8 and 9 suffer from similar issues as above: association of TIL, macrophage regulation, proliferation and TGF-beta response with survival outcomes is well-known and well-studied. Why do we need HPL to rediscover them?”

We would like to point out to reviewer #3 that this comment by reviewer #2 was not about the lack of novel findings. Instead, reviewer #3 seems to believe that HPL tries to rediscover known associations of tumor and TME parameters with survival. We want to clarify again to both reviewers #2 and #3 that this is not the case. Obviously, we do not claim that we discovered that these well-studied biological processes are important for survival. The main point of this analysis is not related to survival at all. What we do show is that HPL-discovered HPCs are correlated

with TILs, macrophage regulation, proliferation and TGF-beta response, i.e. we derive associations between morphology and biological processes, which is where the novelty lies.

Reviewer #3 (Remarks to the Author):

I believe that the authors have addressed most of my points first raised in review at [Redacted]

We thank the reviewer for this comment. In addition, we have implemented DINO and compared its performance on one of HPL's downstream applications.

However and critically, whilst I think the use of self-supervised learning and clustering / community detection is a hot topic and certainly hold promise in the field of histopathology, the applications have to demonstrate utility to the field (e.g. oncology here) but those presented in the manuscript are not convincing, non-novel, and unfortunately make up the majority of the manuscript.

The reviewer's opinion is that the applications of HPL have to demonstrate utility. We can agree with this statement, while emphasizing that utility is not synonymous with novelty. Therefore, we will focus on utility and we will let the editorial team decide on the relevance of novelty. The main application of HPL is the discovery of HPCs. It is obviously useful because it is done in an unbiased and automated way, without relying on (potentially biased) annotations and/or labels from pathologists (**the focus of this study is Pathology, not Oncology**). Building a supervised model would require a large number of annotations by a large number of pathologists. With HPL, this is not necessary. Therefore, **the first application of HPL is to automatically identify these patterns**. However, simply discovering patterns in WSIs is clearly not enough: a series of validations need to be undertaken, and this is indeed a large part of this manuscript. We evaluate the patterns histopathologically and derive associations with molecular phenotypes and outcomes. There are downstream applications as well, most notably the **prediction of recurrence in early-stage lung adenocarcinoma, an unmet need in oncology**. Current recommendations by the IASLC focus on tumor grade, in a study that was led by one of the co-authors (Dr. Moreira). Here we show that an HPC-based model outperforms the grade-based model (see analysis above in response to Reviewer #2) and offer patient-level interpretability through SHAP analysis of the HPC's contribution to the patient's risk of recurrence. We are not sure if the reviewer is an expert in lung adenocarcinoma, but if they are, they would agree that this is a significant step forward. There is no doubt that this model needs to be tested in a clinical trial setting before it can be officially adopted in clinical practice, and this is indeed a future study that we plan to undertake.

My concerns are shared with 2 other reviewers and centre around novelty of HPL applications. For example, finding that TIL macrophages, proliferation or TGF-b response associates with patient survival is a known finding.

We note that the remaining comments from reviewers 1 and 4 are completely unrelated to novelty. Also, reviewer 2's comments are unrelated to overall novelty:

- Comment #4 refers to “usefulness”, which we (through official IASLC guidelines and recommendation) address in our response to that comment.
- Comment #7 refers to why HPL seems to rediscover known associations of molecular phenotypes with survival (restated by reviewer #3 above). We emphasize that HPL does not do that.

It seems that the reviewer feels that we have claimed to discover a link between TILs or macrophages with survival. This is not the case, and we apologize if we did not clearly state this. Instead, what we show is an association between HPCs (histologic patterns) and TILs or macrophages (or other parameters) which are likely to underlie the prognostic power of HPCs, and help to explain their prognostic value. This is in addition to any future novel hypothesis generation which our method will support.

We understand that this reviewer may have had different expectations from SSL models, for example, the discovery of brand new histologic patterns. Indeed, a completely new histologic pattern would be worth publishing at the top journals and making headlines. This was not our goal, and to our knowledge DINO and other approaches have not discovered new pathology either. Perhaps this is not too surprising as tumor histology has been intensively studied for over a century, and so most meaningful histologic patterns have probably already been discovered by humans. Perhaps there are such novel patterns to be found, but we first need to show that SSL can do the task of recapitulating known pathology reasonably well. In this context, we want to clarify that HPL's novelty is the automated discovery of HPCs through SSL, i.e. a set of histomorphologic patterns that are discovered without any (biased) labels or annotations by experts.

Furthermore, HPL can be used to process large collections of unlabeled and unannotated data, effectively converting the enormous complexity and subtlety of histological images into a quantitative vector for subsequent analysis. Merely proposing clusters of learned embeddings is not enough in our opinion, which is exactly what the vast majority of SSL papers have done so far. They focus on proposing patterns and using them for downstream tasks. Our study is different: we propose patterns (HPCs), we ask pathologists to characterize these HPCs (novel), we associate these HPCs with molecular and other parameters (novel, since HPCs are novel), and then we also show that they can be used for downstream tasks, such as predicting recurrence in early-stage lung adenocarcinoma, a true unmet need.

Further novelty comes from the dramatic democratization of histopathology that our method provides. HPL converts H&E images, quite unintelligible to most non-experts, into a map of HPCs with descriptive labels, allowing all users to interpret H&E images in pathological terms with none of the years of training required to interpret these images.

In summary, the novelty does not lie in discovering new links between the TME and clinical outcomes. It is about automating the discovery, characterization and quantification of HPCs to facilitate downstream tasks.

Having a C-index performance of approximately 0.65 is not clinically useful and is below other supervised approaches available.

Our method significantly outperforms the current International Association for the Study of Lung Cancer (IASLC) grades on predicting recurrence-free survival in early-stage LUAD (Moreira et al, J. Thoracic Oncology, 2020). This is a task with high clinical relevance in Pathology.

We now include an analysis to compare the performance of morphological grade and the HPC model at predicting recurrence-free survival (RFS). Grading was provided by Dr. Andre Moreira, the first author of the IASLC recommendation study referenced above.

Of note, in our original RFS analysis we used all 276 patients of the NYU₁ cohort, however only a subset of 256 have an associated grade annotation (Moreira et al. 2020), therefore we only used the subset of 256 patients in the following results.

The HPC model achieves a mean C-index of 0.72 while only morphological grade reaches 0.64. We can conclude that HPC information completely outperforms morphological grade alone when predicting RFS.

Recurrence Free Survival	HPCs	Morphological Grade
C-Index Mean (95% CI)	0.72 (0.67-0.77)	0.64 (0.58- 0.71)

Furthermore, in Table 1, we show that HPL is not inferior to state-of-the-art supervised approaches, namely Porpoise (Chen et al 2022, Cancer Cell) and HIPT (Chen et al 2022 CVPR).

Whilst it's reassuring to have positive controls (i.e. known associations recapitulated) we also would like to see novel inferences previously unexplored.

The novel inferences are the HPCs themselves, and their histological, molecular, and clinical associations. In LUAD, we found 46 HPCs which provide novel granularity compared to the traditionally recognised 6 growth patterns. For example, the solid pattern is split into 5 HPCs characterized by varying immune status. Another novel aspect of our finding is how frequently those HPCs appear in the patient population, which is a crucial discovery and clinical utility. No other supervised approach has reported how frequently do their identified features appear among patients.

Associations between molecular data and the HPCs are entirely novel. While TGF-beta and macrophage are known to be associated with survival, how they are associated with histomorphological patterns is NOT known. We have reported other associations between HPCs

and molecular data. These findings make HPL a timely tool for automatically identifying regions of interest to pursue further spatial biology studies.

One important point I made in my review which was ignored, was that Barlow-twins as a self-supervised model, has been shown to be inferior to other self-supervised methods, namely DINO. For example, the latest Top-1 accuracy for self-supervised methods is DINOv2 (#1 - 86.7%) versus Barlow Twins (#73 - 73.2%). Giving the authors the benefit of the doubt as DINOv2 is a recent development, we can compare Barlow Twins to DINOv1 (#16 - 80.3%). Both Barlow Twins and DINO were published in the same year (2021), so it is unclear to me why the authors would choose an inferior method.

Regarding the Barlow-Twins and DINOv1 comparison, we would like to include some context for our choice:

- The numbers given by the reviewer compare two different deep learning models: ViT DINO1 (80.1%) vs ResNet Barlow Twin (73.2%). When comparing ResNet DINOv1 (75.3%) and ResNet Barlow Twins (73.2%), the difference is 2.1%. They did not compare ViTs (Caron et al 2021 ICCV).
- DINO and Barlow Twins provided a batch size ablation study with a drop of 2% from 1024 to 128 (Figure 2 in Zbontar et al 2021, Table 9 in Caron et al 2021 ICCV). However, the conditions used in DINO's study are different from the SOTA performance. DINO performance is only reported in K-NN (not with a linear classifier) and without multicrop augmentations, the baseline number at batch size 1024 is 59.9% (vs SOTA ResNet50 of 67.5% or ViT-S 74.5%). We aimed to use a self-supervised approach that allowed less computational resources with similar performance to baseline, reducing batch size and less memory consumption without teacher-student approaches. Our approach uses a batch size of 64 and a single Nvidia Titan Xp which can be trained by any user with a small size GPU (12GB).
- A recent study published in CVPR 2023 showed that DINO is not always better than Barlow Twins on histopathology applications (Kang et al 2023 CVPR).

Caron et al. 'Emerging Properties in Self-Supervised Vision Transformers'. ICCV 2021.

Zbontar et al. 'Barlow Twins: Self-Supervised Learning via Redundancy Reduction'. ICML 2021.

Kang et al. 'Benchmarking Self-Supervised Learning on Diverse Pathology Datasets'. CVPR 2023.

We provide the concordance index for LUAD overall survival by replacing Barlow Twins with DINO. We use the SOTA split from Porpoise and HIPT (Chen et al. 2022a and 2022b). Above, we included Barlow Twins figures as a comparison.

DINO LUAD overall survival (resolution 1.5) achieves a c-index of 0.612 with 95% confidence intervals (CI) of 0.569-0.643 in the TCGA Test set and a c-index of 0.597 with 95% confidence intervals (CI) of 0.565-0.625 in the NYU cohort. Barlow Twins (resolution 2.0) achieves a TCGA Test set c-index of 0.600 with 95% confidence intervals (CI) of 0.551-0.649, and an NYU c-index of 0.666 with 95% confidence intervals (CI) of 0.656-0.676.

The performance of both self-supervised methods is similar. However, DINO concordance index fluctuates more with respect to the Leiden resolution parameter (resulting in more/less clusters as explained in the Method) while Barlow Twins remains more stable at different resolutions.

DINO LUAD recurrence-free survival (resolution 1.5) achieves a comparable result to Barlow Twins with a c-index of 0.701 with a 95% confidence interval of 0.642-0.768. In comparison, Barlow Twins (resolution 2.0) reaches a c-index of 0.725 with a 95% confidence interval (CI) of 0.681-0.770.

We have included this analysis in the manuscript as Supplementary Figure 13.

Reviewer #4 (Remarks to the Author):

Regarding point 2 in previous comments, I would still encourage the authors to do a conformal analysis, it is quite a generalized approach and can be easily incorporated into the current setup. That would be a good way to perform the uncertainty quantification.

<https://arxiv.org/abs/2107.07511>

We thank the reviewer for suggesting a specific approach to perform uncertainty quantification. We applied conformal prediction to the LUAD vs LUSC classification model and we present our results below.

We performed label-conditional Conformal Prediction (MICP, Toccaceli et al. 2019) on the LUAD vs LUSC prediction task for a range of coverage (1- α) from 0.05 to 0.95. The figure above shows for LUAD and LUSC the ratios of correct single predictions (A,F), incorrect single predictions (B,G), uncertain predictions (C,H), and empty predictions (D,I). Uncertain predictions are defined as prediction sets that contain both labels and empty predictions are as prediction sets that contain no label. Finally, we include the empirical coverage for LUAD (I) and LUSC (J), highlighting the calibration coverage in the grey dashed line.

We define non-conformity measure as one minus the probability of the correct class. For each coverage measure (1- α), we obtained 1000 samples by running 200 trials per fold, out of the initial folds used in the 5-fold cross-validation LUAD vs LUSC task. For each trial, we randomly subsampled 50% of the total samples on the TCGA validation set (calibration set), TCGA test set (quantification set), and NYU cohort (additional quantification set), resulting into approximately 100 samples for the TCGA validation and test set, and 70 samples for the NYU cohort.

We can see higher uncertainty in TCGA LUAD samples compared to TCGA LUSC samples. The model is also more sensitive to misclassifications in LUAD, barely making any in LUSC. Indeed, this result may be explained by LUAD solid samples, which are harder to differentiate from LUSC than other LUAD subtypes such as acinar, papillary, micropapillary, or lepidic. HPL displays similar confidence on LUAD and LUSC samples of the NYU cohort, with higher uncertainty in LUSC compared to TCGA. Nonetheless, HPL makes larger correct single predictions on both subtypes in the NYU cohort than in TCGA. Possibly due to NYU WSIs being cleaner and less noisy samples than the TCGA counterparts, in particular on the LUAD cases. We have included this analysis in the manuscript as Supplementary Figure 21.

We want to note that for survival, the suggested paper does not include an integration on Cox proportional hazards and survival analysis, because it is not a trivial task. We were only able to find a single quite theoretical paper on this topic: <https://academic.oup.com/jrsssb/article/85/1/24/7008653>. However, it does not provide a direct link to the state-of-the-art concordance index, hazard ratio, and Kaplan-Meier analysis, and we believe it is outside the scope of this work to develop a practical conformal method for Cox survival model.

For immune associations, we have already done False Discovery Rate (FDR) control for that, and it is the best practice in the field. We hope the reviewer would agree that sticking with FDR will help the readers to better compare our findings with existing results on immune associations in the community.

References:

Tocaceli, P., Gammerman, A. 'Combination of inductive mondrian conformal predictors'. Mach Learn 108, 489–510 (2019).

Regarding point 3 in previous comments, you have used TCGA data, which often contains other data modalities (like radiology, genomics) and also patient metadata table (age, sex, etc), where there any for yours? In that case, could you comment or show some analysis on the multi-modal question raised specific to your data instead of a general comment?

This comment is a little unexpected, as we were previously asked for a more general comment: *“This is more of a conceptual point on the same lines, if there are no plans for this at the moment, **you can just answer this theoretically**. How would this method generalize to multi-modal data (say clinical, pathology image, genomics) as the models dealing with large healthcare datasets in the near future would be expected to handle such data.”*

Nevertheless, and even though multimodal integration is outside the scope of the paper, we attempt to address it below:

- Radiology: While we agree this is a very interesting study and would be a nice extension of the solution proposed here, it is a topic on its own which we believe does not fit into this

paper and what the message of this paper is. Also, none of the current authors is an expert in Radiology. In addition, only a very small proportion of the patients in TCGA have matched radiology and pathology images. For our LUAD study, only 20 out of 452 patients (<https://wiki.cancerimagingarchive.net/display/Public/CIP+TCGA+Radiology+Initiative>) have matched radiology and pathology data. It's not at all clear if it is possible to draw any generalizable conclusions from this data. Finally, we currently have no IRB to allow the use of Radiology.

- Molecular data is only available in a subset of the NYU dataset
- Clinical and demographic data was indeed available for both TCGA and NYU. We present the integration of HPCs with this data modality below:

We report a multi-modal LUAD overall survival analysis including HPC patterns and patient metadata (age, sex, and race), patient metadata is included as covariates in the Cox model along with HPC values. We evaluated the model using a 5-fold cross-validation over the TCGA data with a cohort from NYU as an additional independent set. The multi-modal model achieves a mean concordance index (c-index) of 0.636 with 95% confidence intervals (CI) of 0.596-0.680 on the TCGA test set and a mean c-index of 0.650 with 95% CI of 0.636-0.664 on NYU. In addition, we show a SHAP (SHapley Additive exPlanations) plot with top 14 HPCs with the largest impact and patient metadata. Finally, we report high-risk (orange) and low-risk groups (blue) KM plots in TCGA ($p\text{-value } 1.86 \times 10^{-4} < 0.05$) and NYU ($p\text{-value } 3.57 \times 10^{-3} < 0.05$), showing statistically significant separation.

TCGA Cohort

NYU Cohort

We performed the same analysis on multi-modal LUAD recurrence-free survival, using a 5-fold cross-validation over the NYU cohort. The multi-modal model achieves a mean concordance index (c-index) of 0.748 with 95% confidence intervals (CI) of 0.698-0.808. We show a SHAP (SHapley Additive exPlanations) plot with top 14 HPCs with the largest impact and patient metadata and report high-risk (orange) and low-risk groups (blue) in a KM plot, showing statistically significant separation ($p\text{-value } 8.82 \times 10^{-7} < 0.05$). We have included these analyses in the manuscript as Supplementary Figures 14 and 15.

NYU Cohort

REVIEWERS' COMMENTS

Reviewer #3 (Remarks to the Author):

I thank the authors for the systematic answers to my questions and I am fully satisfied with the answers they have provided.

Reviewer #3 (Remarks on code availability):

The code is well documented and well written.

Reviewer #4 (Remarks to the Author):

I am now satisfied with the author responses and feel that the changes in the revised manuscript address the comments raised fully, I am happy for the paper to be progressed to publication from my side if the other reviewers and editors are in agreement.